# Identification and Roles of miR-29b-1-3p and miR29a-3p-Regulated and Non-Regulated lncRNAs in Endocrine-Sensitive and Resistant Breast Cancer Cells

**DOI:** 10.3390/cancers13143530

**Published:** 2021-07-14

**Authors:** Penn Muluhngwi, Carolyn M. Klinge

**Affiliations:** 1Department of Surgery, Feinberg School of Medicine, Northwestern University, Chicago, IL 60611, USA; penn.muluhngwi@northwestern.edu; 2Department of Biochemistry & Molecular Genetics, University of Louisville School of Medicine, Louisville, KY 40292, USA

**Keywords:** miR-29, lncRNA, tamoxifen, endocrine resistance, breast cancer

## Abstract

**Simple Summary:**

Estrogen receptor α (ERα) is a key driver and clinical target in breast cancer, with ~75% of women having ERα+ breast tumors at diagnosis. Endocrine therapies (tamoxifen and aromatase inhibitors) targeting ERα are the preferred treatment for ER+/HER2− breast tumors due to their efficacy and tolerance in most patients. However, patients develop resistance to these endocrine therapies and disease progression to metastasis remains a major clinical problem. There are multiple mechanisms involved in the progression to endocrine resistance, including epigenetic changes in non-coding RNAs that regulate cellular pathways leading to cancer progression and metastasis. This paper summarizes the role of long non-coding RNAs regulated by miR-29 in endocrine-resistant breast cancer.

**Abstract:**

Despite improvements in the treatment of endocrine-resistant metastatic disease using combination therapies in patients with estrogen receptor α (ERα) primary tumors, the mechanisms underlying endocrine resistance remain to be elucidated. Non-coding RNAs (ncRNAs), including microRNAs (miRNA) and long non-coding RNAs (lncRNA), are targets and regulators of cell signaling pathways and their exosomal transport may contribute to metastasis. Previous studies have shown that a low expression of miR-29a-3p and miR-29b-3p is associated with lower overall breast cancer survival before 150 mos. Transient, modest overexpression of miR-29b1-3p or miR-29a-3p inhibited MCF-7 tamoxifen-sensitive and LCC9 tamoxifen-resistant cell proliferation. Here, we identify miR-29b-1/a-regulated and non-regulated differentially expressed lncRNAs in MCF-7 and LCC9 cells using next-generation RNA seq. More lncRNAs were miR-29b-1/a-regulated in LCC9 cells than in MCF-7 cells, including DANCR, GAS5, DSCAM-AS1, SNHG5, and CRND. We examined the roles of miR-29-regulated and differentially expressed lncRNAs in endocrine-resistant breast cancer, including putative and proven targets and expression patterns in survival analysis using the KM Plotter and TCGA databases. This study provides new insights into lncRNAs in endocrine-resistant breast cancer.

## 1. Introduction

The majority of breast tumors express estrogen receptor α (ERα, *ESR1*) [1], which is targeted by selective ER modulators (SERMs, e.g., tamoxifen (TAM)) that competitively inhibit estradiol (E2) and other estrogens from binding and activating ERα’s transcriptional activity and by aromatase inhibitor (AIs, e.g., letrozole) that reduce endogenous estrogen levels [2]. These endocrine therapies have been highly successful in preventing recurrent disease; however, 30%–40% of patients develop resistance to these therapies and have metastatic disease [3,4]. The five-year survival rate for women diagnosed with metastatic breast cancer (mBC) varies between 7.2% and 29% [5]. Survival among women with mBC from ERα-expressing (ER+) primary tumors has increased over time with better therapies—including cyclin-dependent kinase 4 and 6 (CDK4/6) inhibitors (palbociclib/ribociclib/abemaciclib), an mTORC1 (MTOR, mechanistic target of rapamycin kinase) inhibitor (everolimus), and an alpha isoform-specific PI3K inhibitor (alpelisib)—which are used in combination with endocrine therapy (fulvestrant) for ER+ mBC [6,7,8]. Acquired endocrine resistance is the result of multiple mechanisms, including the amplification of growth signaling pathways [9]. Approximately 25%–40% of metastatic tumors in AI-treated breast cancer (BC) patients have been reported to have *ESR1* (ERα) mutations within the ligand binding domain (LBD) [10]. These mutations result in ligand-independent activation of the mutant ERα protein and reduce the efficacy of SERMs and selective ER downregulators (SERDs) fulvestrant, GDC-0810, RU-58688, and AZD9496 [11].

Alterations in the expression of noncoding RNAs (ncRNAs), including circular RNA (circRNA) [12,13,14], microRNA (miRNA) [15,16], and long noncoding RNAs (lncRNAs) [17,18] have been reported in breast tumors and in circulation in BC patients, with specific alterations in endocrine resistance [19,20,21,22]. miRNAs and lncRNAs are epigenetic regulators of human cancers [23]. Pre-miRNAs and lncRNAs are post-transcriptionally modified, e.g., by methylation on N6 of adenosine (m6A), which alters the processing and interaction with RNA binding proteins; thus, epitranscriptomic modification regulates cellular events in BC and in other cancers [24,25,26]. miRNAs regulate mRNA translation and RNA stability by base-pairing between the seed sequences at 5′ positions 2–7 or 2–8 of the miRNA, with ~7 bp miRNA recognition elements (MREs) in the 3′ UTR of their target mRNAs within the RNA-induced silencing complex (RISC) [27]. The current miRBase database (release 22.1) contains 2654 mature human miRNAs http://www.mirbase.org/ (accessed on 25 April 2021) [28]. Depending on the tissue, miR-29 family members (miR-29a (MI0000087), miR-29b-1 (MI0000105), miR-29b-2 (MI0000107), and miR-29c (MI0000735) act an oncomiRs or as tumor suppressor miRNAs [29,30,31,32,33,34]. We previously reported that a low expression of miR-29a-3p and miR-29b-3p is associated with lower overall BC survival before 150 mos and that transient, modest overexpression of miR-29b1-3p or miR-29a-3p inhibited MCF-7 tamoxifen (TAM)-sensitive and LCC9 TAM-resistant BC cell proliferation [35]. The TAM- and fulvestrant-resistant LCC9 cell line was derived from MCF-7 tumor xenografts in TAM-treated mice and is ER+ [36,37]. We attributed this observation in part to the repression of the transcription of ATP synthase subunit genes *ATP5G1* and *ATPIF1* by miR-29b-1-3p and miR-29a-3p [35]. miR-29b1-3p and miR-29a-3p are derived from the same precursor-miRNA (pre-miRNA) from chromosome 7, whereas miR-29b-2 and miR-29c are located on chromosome 1 [34]. A recent study confirmed reduced miR-29a-3p to be a disease-specific survival prognostic indicator in BC [38]. miR-29 family members (miR-29a, b-1, b-2, and c) also target additional genes, i.e., *ADAM12, ANGPTL4, ARP1B1, DICER1, TTP, PTEN, KLF4, MYP, LOX, MMP, PDFGC, SERPINH1,* and *VEGFA,* (reviewed in [20,39]).

The current GENECODE (version 384) of the human genome includes 60,649 genes, 16,888 long noncoding RNAs (lncRNAs), and 1879 miRNAs https://www.gencodegenes.org/human/stats.html (accessed on 25 April 2021). The function of most lncRNAs remains to be characterized. By definition, lncRNAs are ncRNAs > 200 nucleotides in length [40]. lncRNAs are transcribed by RNA pol II from intergenic (lincRNA), intronic, antisense (AS), and regions overlapping mRNAs from loci marked with H3K4me3 at the promoter and H3K36me throughout the transcript body (reviewed in [41,42]). LncRNAs include enhancer RNAs (eRNAs), promoter upstream transcripts (PROMPTs), and small nucleolar RNA (snoRNA)-ended lncRNAs (sno-lncRNAs) [43]. LncRNAs are found in a low abundance in part due to their rapid degradation by the RNA exosome [44]. Most lncRNAs are nuclear, but lncRNAs have functional roles in the cytoplasm, tethered to cell membranes, in mitochondria, and are sorted into exosomes for systemic distribution, which contributes to metastasis (reviewed in [41]). In the cytoplasm, lncRNAs bind RNA-binding proteins (RBPs) and can positively or negatively affect translation through their interaction with translation factors and ribosomes [43]. LncRNAs can act in *trans* to regulate genes or other transcripts at a distance or in *cis* to regulate neighboring genes [42]. Photoactivatable-ribonucleoside-enhanced crosslinking and immunoprecipitation (PAR-CLIP) studies identified miRNA–lncRNA interactions clustering in the mid-regions and 3′ ends of lncRNAs [45]. LncRNAs act as ‘sponges’ for miRNA by acting as competing endogenous RNA (ceRNA), thus blocking the repressive activity of miRNAs, i.e., blocking miRNA binding to the 3′ UTR of their target transcripts [21]. The loss of repression leads to increased target mRNA translation and protein abundance. miRNA–lncRNA interaction also regulates lncRNA stability [46]. A network analysis of ncRNAs in cancer drug resistance associated lncRNAs, miRNAs, and TAM resistance, including lncRNAs *MALAT1* and *CCAT2*; miR-221, miR-222, miR-26a, miR-29a, and miR-29b (of which the isoform was not specified), has been described [47]. lncRNAs can also act as intracellular scaffolds, e.g., *HOTAIR* provides a platform for PRC2 and LSD1 histone-modifying complexes to promote H3K27 methylation and H3K4 demethylation to silence genes and promote metastasis in BC [48].

Transcriptomic regulation of endocrine resistance in BC cells involves regulatory networks of lncRNAs, miRNAs, circRNAs, and mRNAs [49]. In this study, we identified the lncRNAs regulated by miR-29b-1-3p and miR-29a-3p in MCF-7 and LCC9 breast cancer cells. In addition, we identified the lncRNAs that are differentially expressed in the two cell lines independently of miR-29b-1 or miR-29a regulation. We review the roles of these lncRNAs and their targets in BC progression, endocrine therapy responses, and metastasis.

## 2. Materials and Methods

### 2.1. RNA Sequencing

RNA sequencing was previously described in [35]. In brief, single read sequencing (75–76 cycles) was performed using the 500 High-Output v2 (75 cycle) sequencing kit (Illumina, Foster City, CA, USA) on an Illumina NextSeq500 instrument. Obtained read sequences were mapped to the human reference genome version GRCh37.1 using the mapping algorithm Tophat version 2.0.2 (Toronto, ONT, Canada) Using Cufflinks version 2.2.1 (Seattle, WA, USA) and annotations found at ENSEMBL, *Homo sapiens* GRCh37.73.gtf expression levels at loci were quantified. Data are from the GEO database: accession number GSE81620.

### 2.2. In Silico Pathway and Network Analysis

Data from RNA-seq were analyzed such that transcripts selected had a log 2 fold-change greater than 0.34 (or −0.34 for repressed transcripts) and a statistically significant threshold q-value less than 0.05. lncRNA–miR-29 interactions were checked against those verified and predicted using DIANA-LncBase v3 [50] (Athens, Greece). Network and pathway enrichment analysis for the lncRNAs was evaluated using a fee-based site license for the web-based software MetaCore version 21.1 (Cortellis, Philadelphia, PA, USA) https://portal.genego.com/ (accessed on 25 April 2021). MetaCore is a manually curated database of experimental findings and interactions [35,51,52].

## 3. Results and Discussion

### 3.1. Identification of miR-29b-1/a-Downregulated lncRNAs and Their Roles in Breast Cancer

To identify miR-29b-1-3p- and miR-29a-3p-regulated transcripts in MCF-7 TAM-sensitive and LCC9 TAM-resistant BC cells and their possible roles in endocrine-resistant BC progression and metastasis, we previously transfected each cell line with either pre-miR™ negative control, pre-miR-29b-1-3p, pre-miR-29a-3p, anti-miR negative control, or anti-miR-29 individually or in combination. By carrying out the co-transfection of each cell line with pre-miR-29b-1-3p + anti-miR-29 and pre-miR-29a-3p + anti-miR-29 and comparing the resulting transcriptomes to those with controls and those transfected with pre-miR-29b-1-3p or pre-miR-29a-3p, we identified miR-29b-1-3p- and miR-29b-1-3p-regulated transcripts, respectively (Figure 1). For example, for an lncRNA downregulated by miR-29b-1-3p, we expect a decrease in the fragments per kilobase of transcript per million mapped reads (FPKM) value of that lncRNA in cells transfected with pre-miR-29b-1-3p and an increase in the FPKM of that lncRNA in cells transfected with pre-miR-29b-1-3p + anti-miR-29 (Table 1). Conversely, for an lncRNA upregulated by miR-29b-1-3p, we expect an increase in the FPKM of that lncRNA in cells transfected with pre-miR-29b-1-3p and a decrease in the FPKM of that lncRNA in cells transfected with pre-miR-29b-1-3p + anti-miR-29 (Table 2).

We identified 19 lncRNAs that were upregulated by anti-miR-29 in LCC9 cells and/or MCF-7 cells with pre-miR-29b-1/a + anti-miR-29 transfection (Table 1), suggesting miR-29 regulation. As described previously in our identification of differentially expressed genes (DEGs, i.e., mRNAs) regulated by miR-29b-1-3p and miR29a-3p (miR-29b-1/a) [35], we observed more lncRNAs downregulated by miR-29b-1/a in LCC9 than in MCF-7 cells (18 and two, respectively) with only *TUG1* being downregulated by miR-29b-1/a in both cell lines. Fourteen miR-29b-1/a-downregulated lncRNAs were more highly expressed in MCF-7 cells than in LCC9 cells: *SOX2-OT, LINC00473, MIR17HG, FIRRE, DLEU1, OIP5-AS1, JPX, DANCR, CYTOR, TUG1, SNHG8, SNHG5, DSCAM-AS1*, and *GAS5* (Table 1). Two miR-29b-1/a-downregulated lncRNAs were more highly expressed in LCC9 than in MCF-7 cells: LINC00221 and MIR99HG (Table 1). We used DIANA-LncBase v3.0 (Athens, Greece) [50] to examine lncRNA-miR-29b-1/a interaction and found that 10 of the 19 putative interactions identified in MCF-7 and/or LCC9 cells have been experimentally validated (Table 1). MetaCore enrichment analysis identified gene ontology (GO) processes, which are shown in Appendix A, but these only included the lncRNA *TUG1*. MetaCore network analysis is shown in Appendix A and selected networks are shown in Figure 2, Figure 3 and Figure 4.

### 3.2. Identification and Functional Roles of lncRNAs Downregulated by miR-29b-1/a

The abundance of *LINC00511* and *TUG1* was increased by anti-miR-29 in both MCF-7 and LCC9 cell lines transfected with either pre-miR-29a-3p or pre-miR-29b-1-3p, suggesting the downregulation of these lncRNAs by miR-29b-1/a in these cells (Table 1). The abundance of *SOX2-OT* and *MIR17HG* was increased by anti-miR-29 only in MCF-7 cells transfected with either pre-miR-29a-3p or pre-miR-29b-1-3p, suggesting the selective downregulation of these lncRNAs by miR-29b-1/a in MCF-7 cells (Table 1). The abundance of *LINC00221, LINC00473, FIRRE, MIR99AHG, DLEU1, APTR, OIP5-AS1, JPX, DANCR, CYTOR, CRNDE, SNHG8*, *SNHG5, DSCAM-AS1*, and *GAS5* was increased by anti-miR-29 only in LCC9 cells transfected with either pre-miR-29a-3p or pre-miR-29b-1-3p, suggesting the selective downregulation of these lncRNAs by miR-29b-1/a in LCC9 TAM-resistant BC cells (Table 1).

*LINC00221* was not detected in MCF-7 cells but was increased by the anti-miR-29 transfection of LCC9 cells transfected with pre-miR-29b-1-3p or pre-miR-29a-3p, suggesting that miR-29b-1-3p and miR-29a-3p selectively downregulate *LINC00221* in LCC9 TAM-resistant BC cells. In contrast to our findings, a previous study observed higher *LINC00221* in MCF-7 compared to another TAM-resistant cell line derived from MCF-7 cells (LCC2) and reported that the higher expression of *LINC00221* in ER+ BC patients was associated with a higher probability of survival [49]. The reason for this difference may be methodological: the previous study used the Agilent human lncRNA + mRNA Array V4.0 for the profiling of lncRNAs and mRNAs in MCF-7, LCC2, and LCC9 cells [49], whereas we used direct RNA seqs [35].

A report profiling Ago2:RNA interactions using HITS-CLIP in human post-mortem brain tissue identified an *SOX2-OT*–miR-29b-1-3p interaction [53]. An analysis of TCGA breast tumor cells identified the amplification of *SOX2-OT* as a putative lncRNA driver of BC [54]. SOX2-OT was identified in network 3 with *DLEU1, CYTOR* (*LINC00152*), and *TUB1* (Figure 3). The reduction of the *SOX2-OT* abundance by miR-29b-1-3p in MCF-7 cells fits with its ‘anti-tumorigenic’ activity in BC [55].

*LINC00473* was higher in breast tumors and BC cell lines compared to normal breast tissue and breast epithelial cells [56]. High *LINC00473* expression was correlated with lymph node (LN) metastasis, clinical stage, and poor outcomes in BC patients [56]. In MCF-7 and MDA-MB-231 cells, *LINC00473* is a ceRNA for miR-497 [56] and for miR-198 [57]. *LINC00473* increases *CCND1* transcription in MCF-7 cells by increasing the recruitment of pCREB and H3K27ac to activate the promoter, thus increasing cell proliferation [58]. The reduction of the *LINC00473* abundance by miR-29a-3p and miR-29b-1-3p fits with their ‘anti-tumorigenic’ activity in BC (reviewed in [30]).

*MIR17HG* (miR-17-92a-1 cluster host gene) encodes six miRNAs: *MIR17, MIR18A, MIR19A, MIR20A, MIR19B1,* and *MIR92A1*, which are members of four seed families (miR-7, miR-18, miR-19, and miR-92) [59]. The miR-17-92a cluster is considered oncogenic (reviewed in [60]). miR-18a directly targets *ESR1* and reduces ERα in MCF-7 and BT-474 cells [61]. miR-17-5p targets pro-metastatic genes involved in transforming growth factor β (TGFβ) and hypoxia signaling in basal-like BC [62]. *MIR17HG* was included in network 7 with β-catenin (Appendix A). The reduction of *MIR17HG* abundance by miR-29a-3p and miR-29b-1-3p fits with their ‘anti-tumorigenic’ activity in BC (reviewed in [30]).

*FIRRE* forms “RNA clouds” in the nucleus, binds the nuclear matrix protein HNRNPU (heterogeneous nuclear ribonucleoprotein U), and serves as a platform for trans-chromosomal associations [63]. *FIRRE* is transcribed from the active X chromosome and acts in *trans* and *cis* to maintain X chromosome inactivation [64]. No reports of FIRRE/LNC01200 in BC were located in PubMed.

*MIR99AHG* (LINC00478, miR-99a-Let-7c cluster host gene) abundance was higher in LCC9 cells compared to MCF-7 cells and its abundance was significantly increased by anti-miR-29 in LCC9 cells transfected with pre-miR-29a-3p or pre-miR-29b-1-3p (Table 1). *MIR99AHG* (chromosome 21q21.1) is transcribed as a polycistronic primary transcript that produces a spliced lncRNA and three intronic miRNAs: miR-99a, miR-125b-s, and Let-7c [66]. The miRNA miR-99a/let-7c/miR-125b-2 cluster is upregulated in luminal A compared with luminal B human breast tumors [67]. *MIR99AHG* was identified in network 4 (Appendix A) with *DLEU1* (Figure 4) to be regulated by FOX3P, but no publications were identified on the role of FOX3P in BC in PubMed.

*LINC00511* was demonstrated to directly bind miR-29c-3p, reducing its expression in MCF-7 cells, and relieving its repression of *CDK6* [68]. Previous studies demonstrated an oncogenic role for *LINC00511* that was highly upregulated in breast tumors and BC cell lines, notably in TNBC [69]. *LINC00511* acts as a ceRNA for miR-185-3p [70] and for miR-150 [71] in BC cells. ERα deficiency was reported to increase LINC00511 in breast tumors and RNA IP experiments demonstrated that *LINC00511* interacts with *EZH2* in UACC-812 and MDA-MB-231 TNBC cells [72]. A recent study reported that blocking *LINC00511* using cell-penetrating peptide (CPP)-loaded nanobubbles (CNBs) loaded with siLINC00511 was shown to inhibit MDA-MB-231 cell growth and to enhance sensitivity to cisplatin in vitro [73]. The reduction of *LINC00511* abundance by miR-29a-3p and miR-29b-1-3p fits with their ‘anti-tumorigenic’ activity in BC (reviewed in [30]).

*DLEU1* abundance was significantly increased in LCC9 cells transfected with anti-miR-29 and either pre-miR-29a-3p or pre-miR-29b-1-3p (Table 1), suggesting the downregulation of *DLEU1* by miR-29b-1/a, a result confirmed by DIANA-LncBase v3 (Athens, Greece) [50]. *DLEU1* expression was higher in breast tumors than in normal breast tissue [74]. *DLEU1* was identified in network 4 (Appendix A) with *MIR99AHG* (Figure 4). The network shows the interaction of *FOXP3* with *DLEU1* and *MIR99AHG*. *FOXP3* functions as a tumor suppressor in BC [75]. Further studies are needed to explore the role of *DLEU1* in TAM-resistant BC.

Bioinformatic analysis identified ten miRNAs as putative *APRT* interactors, but miR-29 was not included [76]. *APTR* was demonstrated to repress *CDKN1A* transcription by binding to the promoter and recruiting the PRC2 complex [77]. *APTR* was recently reported to be increased in breast tumors compared to normal adjacent breast tissue and to be higher in larger tumors, suggesting an oncogenic function [76]. Another recent study reported that *APTR* directly interacts with ERα in Ht-UtLM human leiomyoma cells [78]. *APTR* was identified in network 2 (Appendix A) with *TUG1, CRNDE, DANCR*, and *SNHG5* (Figure 2). The network shows the interaction of *APRT, SNHG5,* and *TUG1* with miR-132-3p. miR-132 was identified as a tumor suppressor in BC [79]. Additional studies are needed to parse the role of *APTR* in BC.

RNA immunoprecipitation (RIP) and small RNA-seq identified miR-29a, b, and c as interacting directly with *OIP5-AS1* in HeLa cells [80]. The authors demonstrated that *OIP5-AS1* functions to decrease target mRNA abundance while increasing target miRNA levels [80]. *OIP5-AS1* was higher in breast tumors compared with normal breast tissue and high *OIP5-AS1* correlated with tumor size, LN metastasis, and tumor grade [81]. Another study found that *OIP5-AS1* expression correlates with a high risk of worse outcomes for luminal BC patients [82]. Knockdown of *OIP5-AS1* in MDA-MB-231 cells inhibited xenograft tumor growth in BALB/c nude mice, validating its oncogenic activity in vivo [81]. *OIP5-AS1* acts as ‘sponge’ for RNA-binding protein HuR in HeLa cells, keeping HuR from interacting with mRNAs [83]. HuR is elevated in breast tumors compared to normal breast and increases the stability of a number of regulatory transcripts including *ESR1, STAT3, ERBB*2, and *FOXO1* to stimulate cell proliferation, invasion, and migration [84]. *OIP5-AS* triggers target-directed miRNA degradation (TDMD) of miR-7 in human cell line K562 [85]. Other studies have demonstrated that *OIP5-AS*1 acts as a ceRNA for miR-340-5p, which normally targets and downregulates *ZEB2* [86] and for miR-216a-5p [87] in BC cells. The reduction of *OIP5-AS1* abundance by miR-29a-3p and miR-29b-1-3p fits with their ‘anti-tumorigenic’ activity in BC (reviewed in [30]).

*JPX* positively regulates *XIST* promoter activity by binding *CTCF* (a transcription factor) and repressing its binding to the XIST promoter [88]. *XIST* and *JPX* expression is reduced in breast tumors and BC cell lines due to hypermethylation [89]. The decrease in *JPX* by miR-29b-1/a in LCC9 cells may relate to miR-29b-1/a’s anti-proliferative activity in this cell line [19,20,35,39].

*DANCR* was reported to be higher in basal-like than luminal breast tumors [90]. A high *DANCR* level is associated with reduced overall survival [OS] in TNBC patients [91]. Knockdown of *DANCR* suppressed MDA-MB-231 and MDA-MB-468 TNBC cell proliferation and xenograft tumor growth in mice [91]. *DANCR* interacts directly with the retinoid X receptor alpha (*RXRA*, RXRα) protein and increased its serine 49/78 phosphorylation via GSK3β, resulting in increased *PIK3CA* transcription and activation of the PI3K/AKT pathway in TNBC [91]. *DANCR* is a ceRNA for miR-4319, upregulating VAMP-associated protein B and C (*VAPB*) [92]. *TUFT1* (tuftelin 1) increased *DANCR* expression [93]. *DANCR* is also a ceRNA for miR-874-3p, resulting in de-repression of SOX2 and stimulating epithelial–mesenchymal transition (EMT) in TNBC [93]. A recent study reported that *DANCR* promoted the binding of EZH2 to the promoter of *SOCS3*, thus repressing S*OCS3* to promote EMT, inflammation, and BC stem cells (BCSC) [94]. *DANCR* was identified in network 2 (Appendix A) with *TUG1, CRNDE, APTR*, and *SNHG5* (Figure 2). Figure 2 indicates that *DANCR* and *CRNDE* interact with miR-33a-5p, which is downregulated in breast tumors and acts as a tumor suppressor [95]. Indeed, *DANCR* is a ceRNA for miR-33a-5p in pancreatic beta cells [96], osteosarcoma cells [97], and other cancer cell types, but no reports were found for this interaction in breast tumors or BC cell lines. Further experiments are needed to determine whether *DANCR* is a ceRNA for miR-33a-5p in BC.

The abundance of lncRNA *CYTOR* (LINC00152) was higher in MCF-7 than LCC9 cells (Table 1). These findings are in contrast to a report showing higher *CYTOR* in two other TAM-resistant MCF-7 cell lines compared to the parental MCF-7 cells [98]. *CYTOR* was shown to be a ceRNA for miR-125-5p, resulting in increased serum response factor (*SRF*) and activated Hippo and MAPK signaling pathways in the TAM-R cell lines [98]. *CYTOR* was elevated in breast tumors and in plasma from BC patients compared to normal controls [99]. Higher *CYTOR* was associated with reduced OS in a study of 70 breast tumors [100]. This study identified an interaction between *CYTOR* and KLF5 in MDA-MB-231 and MCF-7 cells that stabilized KLF5 (Kruppel-like factor 5) protein and enhanced tumorigenesis. The authors also demonstrated that KLF5 binds the *CYTOR* promoter and increases *CYTOR* transcription [100]. *CYTOR (LINC00152)* was identified in network 3 (Appendix A) with *DLEU7-AS1, GAS5, SOX2OT,* and *TUG1* (Figure 3). That network indicates that YY1 (YY1 transcription factor) increases *CYTOR* (LINC00152) expression. YY1 correlates with HER2/ERBB2 expression in breast tumors [101] and is upregulated by NFκB signaling and stimulates the expression of BC stem cell (BCSC) transcription factors OCT4, SOC2, and NANOG [102]. While there was no difference in YY1 transcript levels between MCF-7 and LCC9 cells, YY1 was downregulated by miR-29b-1/a in LCC9, but not MCF-7 cells [35]. In summary, the downregulation of *CYTOR* by miR-29b-1/a in LCC9 cells fits with their anti-proliferative activity in this cell line [19,20,35,39].

*CRNDE* expression is higher in breast tumors than in normal breast tissue and was associated with larger tumor size, advanced tumor, nodes, and metastases (TNM) stage and was correlated with reduced OS [103]. *CRNDE* is a ceRNA for miR-136, resulting in activation of WNT/β-catenin signaling in MDA-MB-231 cells [103]. Wnt signaling stimulates BCSC adhesion, proliferation, and invasion to promote metastasis [104]. IGF/insulin signaling represses CRNDE [105]. *CRNDE* was identified in network 2 (Appendix A) with *TUG1, APTR, DANCR*, and *SNHG5* and was indicated, along with *DANCR*, to interact with miR-33a-5p (Figure 2). *CRNDE* was reported to be a ceRNA for miR-33a-5p in hepatocellular carcinoma (HCC) [106].

In agreement with the data presented in Table 1, *TUG1* directly binds and reduces the expression of miR-29b and miR-29c [107]. The expression of *TUG1* is higher in breast tumors and cell lines compared to normal breast tissue [108]. *TUG1* expression was higher in HER2-enriched and basal-like breast tumor subtypes compared to luminal A [109]. Knockdown of *TUG1* reduced BC cell proliferation and xenograft tumor growth in vivo by increasing miR-9, resulting in a reduction of miR-9 target *MTHFD2* [108]. *TUG1* expression is associated with doxorubicin (Dox)-resistance in BC and *TUG1* interaction with miR-9-5p increases translation factor eIF5A2 [110]. *TUG1* is also a ceRNA for miR-197 in TNBC cell lines MDA-MB-231 and BT549, thus increasing nemo-like kinase (*NLK*) expression and resulting in enhanced cisplatin resistance [111]. *TUG1* was identified in networks 2 and 3 (Appendix A) with *CRNDE, APTR, DANCR, SNHG5,* and with *DLEU7*-*AS1, GAS5, SOX2OT,* and *LINC00152* (Figure 2 and Figure 3). *TUG1* was shown to interact with *BRM* (SWI/SNF-related, matrix-associated, actin-dependent regulator of chromatin, subfamily A, member 2), *CRYM* (crystallin Mu), and *LCE1D* (late cornified envelope 1D) in network 2 (Figure 2) and with *UGT2B7* (UDP glucuronosyltransferase family 2 member B7) which has unique specificity for 3,4-catechol estrogens and estriol, suggesting that TUG1 may reduce active estrogens [112]. *TUG1* polymorphisms are associated with BC responses to systemic therapy and responses [112]. The reduction of *TUG1* abundance by miR-29a-3p and miR-29b-1-3p fits with their ‘anti-tumorigenic’ activity in BC (reviewed in [30]).

*SNHG8* expression was higher in breast tumors and BC cell lines compared to normal breast tissue and cell lines, respectively [113]. *SNHG8* was reported to be a ceRNA for miR-634, relieving the repression of *ZBTB20* [113,114] and for miR-384, relieving the repression of HDGF [115] in BC. Additional studies are needed to determine the mechanisms by which miR-29b-1/a regulate *SNHG8* abundance in LCC9 cells.

*SNHG5* was more highly expressed in MDA-MB-231 cells than in MCF-7 cells [116]. Another study reported high levels of *SNHG5* in SK-BR3 HER2+ BC cells [117]. *SNHG5* was reported to be higher in TNBC compared to luminal A or B BC cell lines and to be a ceRNA for miR-154-5p, thus relieving the repression of *PCNA* and upregulating cell proliferation [118]. *SNHG5* was identified in network 2 (Appendix A) with *TUG1, CRNDE, APTR,* and *DANCR* and depicted as regulating miR-132-3p with *APTR* and *TUG1* (Figure 2).

Although it is more highly expressed in MCF-7 cells, the abundance of *DSCAM-AS1* was significantly increased by anti-miR-29 only in LCC9 cells transfected with either pre-miR-29a-3p or pre-miR-29b-1-3p, suggesting that *DSCAM-AS1* is regulated by miR-29b-1/a selectively in LCC9 cells (Table 1). *DSCAM-AS1* was identified as the lncRNA that was most upregulated by “apo-ERα” (non-ligand-occupied ERα) in a bioinformatics analysis of BC cell lines and tumor tissues [119]. That study showed that luminal A and B breast tumors had higher *DSCAM-AS1* expression compared to normal breast tissue and HER2+, or basal-like breast tumors. In follow-up experiments in MCF-7 cells, knockdown of *DSCAM-AS1* increased markers of EMT [119]. A bioinformatics interrogation of 947 breast tumor RNA-seq libraries identified gene sets positively correlated with *DSCAM-AS1* expression as being significantly associated with clinical signatures of cancer aggression, TAM resistance, as well as a higher grade, stage and metastasis [120]. A more recent report confirmed the highest levels of *DSCAM-AS1* in luminal B breast tumors and that *DSCAM-AS1* expression is correlated with disease relapse [121]. Interestingly *DSCAM-AS1* interacts with the RNA binding protein hnRNPL (*HNRNPL,* heterogeneous nuclear ribonucleoprotein) and influenced alternative splicing in MCF-7 cells [121]. Knockdown of *DSCAM-AS1* resulted in decreased expression of many cell-cycle-related genes, including *MYC, RET, TOP2A*, and *POL2A* in MCF-7 cells, implicating it as a driver of cell proliferation in BC [121]. Knockdown of *DSCAM-AS1* was reported to enhance the inhibitory activity of TAM in a TAM-resistant MCF-7 cell line that had higher *DSCAM-AS1* expression compared to parental MCF-7 cells [122]. As indicated in Table 1, we observed a lower *DSCAM-AS1* abundance in LCC9 TAM-resistant cells compared to parental MCF-7 cells. Differences in the derivation of TAM-resistant MCF-7 cells and their culture conditions likely contributed to this difference. *DSCAM-AS1* was reported to be a ceRNA for miR-137, relieving the repression of *EPS8* (epidermal growth factor receptor pathway substrate 8), which simulates MCF-7 growth in vitro and as tumor xenografts in vivo [122].

*GAS5* abundance was ~2.4-fold higher in MCF-7 than in LCC9 cells (Table 1). Others have also reported that *GAS5* expression is lower in TAM-resistant MCF-7 cells compared to MCF-7 cells [123]. *GAS5* levels were significantly increased by anti-miR-29 only in LCC9 cells transfected with either pre-miR-29a-3p or pre-miR-29b-1-3p, suggesting that *GAS5* is downregulated by miR-29b-1/a selectively in LCC9 cells (Table 1). *GAS5* is considered to be a tumor suppressor that inhibits cell proliferation and stimulates apoptosis in BC cells [124]. *GAS5* transcript levels are lower in breast ductal carcinomas compared to adjacent normal breast tissue [125]. *GAS5* was identified in plasma from BC patients, but due to low levels of lncRNAs, *GAS5* was not considered to be prognostic in that study [126]. *GAS5* is a ceRNA for oncogenic miR-21 [127]. A network analysis of ncRNAs in the trastuzumab-resistance-associated lncRNAs *GAS5*, miR-16, and miR-155 has been reported [47]. *GAS5* expression was low in breast tumors from trastuzumab-treated patients [128]. miR-21 negatively regulates *GAS5* [129]. *GAS5* is a ceRNA for miR-222 in TAM-resistant BC cells, thus upregulating tumor suppressor PTEN (phosphatase And tensin homolog) [123]. *GAS5* was identified in network 3 with *DLEU1, SOX2-OT, CYTOR* (LINC00152), and *TUB1* (Figure 3) and was depicted as interacting with P53. Although a role for *GAS5* in regulating P53 in BC is unknown, this association has been detected in other cancers, e.g., neuroblastoma (reviewed in [130]). The reduction of *GAS5* abundance by miR-29a-3p and miR-29b-1-3p in LCC9 fits with their ‘anti-tumorigenic’ activity in BC (reviewed in [30]).

### 3.3. Identification of miR-29b-1/a-Upregulated lncRNAs and Their Roles in Breast Cancer

We identified nine lncRNAs that were upregulated in response to the transfection of MCF-7 and/or LCC9 cells with pre-miR-29b-1-3p or pre-miR-29a-3p and downregulated in response to the co-transfection with anti-miR-29 (Table 2), suggesting that miR-29b-1/a-mediated upregulation. The abundance of three lncRNAs—*BCYRN1, NALT1*, and *NEAT1*—was increased by both miR-29b-1-3p and miR-29a-3p in both cell lines. Six of the miR-29b-1/a-upregulated lncRNAs were more highly expressed in MCF-7 cells than in LCC9 cells and two upregulated lncRNAs were more highly expressed in LCC9 cells than in MCF-7 cells (Table 2). MetaCore pathway analysis identified one pathway associated with lncRNA *UCA1*: development: YAP (*YAP*, yes 1-associated transcriptional regulator)/TAZ (*TAFAZZIN,* tafazzin, phospholipid-lysophospholipid transacylase)-mediated co-regulation of transcription (Figure 5). MetaCore enrichment analysis identified GO processes that are shown in Appendix A, but these only included the lncRNA *MALAT1*. The MetaCore network analysis is shown in Appendix A and selected networks are shown in Figure 6 and Figure 7. An increase in the abundance of an lncRNA by miRNA may result from the miRNA-mediated reduction of a transcription factor that normally increases that lncRNA’s transcription or by the reduction of a factor, e.g., an RNA binding protein, that decreases lncRNA stability. The expression of *BCYRN1, UCA1, ABHD11-AS1, NALT1*, and *NEAT1* was decreased by anti-miR-29 in both cell lines transfected with either pre-miR-29a or pre-miR-29b-1-3p, suggesting the upregulation of these lncRNAs by miR-29b-1/a in these cells (Table 2). The expression of *DGCR5, TINCR,* and *MALAT1* was reduced by anti-miR-29a-3p only in LCC9 cells transfected with either pre-miR-29a-3p or pre-miR-29b-1-3p, suggesting the selective upregulation of these lncRNAs by miR-29b-1/a in LCC9 TAM-resistant BC cells (Table 2).

*DGCR5* is upregulated in some cancers, e.g., lung and gallbladder cancers, but is low in many other cancers, including HCC, ovarian, cervical, pancreatic, and thyroid (reviewed in [131]). *DGCR5* was higher in TNBC tumors compared to normal breast tissue [132]. Further experiments are needed to evaluate the role of *DGCR5* in BC and endocrine resistance.

Knockdown of *BCYRN1* reduced the viability and stimulated apoptosis of MCF-10A ‘normal’ breast epithelial cells and MCF-7, MDA-MB-231, SK-BR-3, and T47D BC cell lines [133]. A recent study demonstrated that *BCYRN1* knockdown reduced translation, whereas stable overexpressed *BCYRN1* was associated with polysomes and enhanced translation, but reduced MCF-7 cell growth [133]. These reports suggest that *BCYRN1* may be oncogenic in BC, but further studies are needed to determine the role and targets of *BCYRN1* in BC and the mechanism by which miR-29b-1/a increase *BCYRN1* abundance in MCF-7 and LCC9 cells.

*UCA1* was not predicted to interact with miR-29 in DIANA-LncBase [50]. A previous report showed higher levels of *UCA1* in LCC2 and LCC9 TAM-resistant cell lines compared to MCF-7, and levels were comparable to those in BT474 HER2+ BC cells [134]. Isolated exosomes from TAM-resistant LCC2 BC cells contained ~25-fold higher *UCA1* levels compared to parental MCF-7 cells and the incubation of MCF-7 cells with exosomes from LCC2 cells resulted in decreased growth inhibition by TAM [135], although no uptake of *UCA1* or other lncRNAs or miRNAs was observed. *UCA1* was upregulated in MCF-7 cells with TAM treatment and is a ceRNA miR-18a, resulting in increased HIF1α, which increases *UCA1* expression [134]. *UCA1* is also upregulated in trastuzumab and paclitaxel resistance (reviewed in [136]) and in DOX-resistant MCF-7 cells [137]. UCA1 expression was upregulated in breast tumors compared to normal breast tissue and stabilized by its interaction with hnRNP I [137]. *UCA1* expression was associated with LN metastasis in breast tumors and reduced OS in BC patients [138]. On the other hand, a recent review found that reduced *UCA1* was a poor prognostic biomarker of luminal BC by controlling the tumor necrosis factor (TNF) signaling and immune responses [139]. *UCA1* transcription is directly upregulated by TGFβ-activated TEAD1 (TEA domain transcription factor 1) and by SMAD2/3 recruitment to the UCA1 promoter in BC cells [140]. *UCA1* was identified in network 1 with NEAT1, *MALAT1, TINCR*, and *SMAD2* (Figure 6) and, in agreement with the previous citation, was depicted as being stimulated by SMAD2. UCA1 was shown to repress miR-129-5p (Figure 6), which targets the 3′ UTR of *FMR1* (fragile X mental retardation protein (FMRP)), an RNA-binding protein [141]. *UCA1* is a ceRNA for miR-129, thus upregulating *SOC4* in renal cell carcinoma (RCC) [142]. Likewise, *UCA1* repression of miR-129 increased *ABCB1* in ovarian cancer cells [143]. There are no reports on this interaction in BC cells; however, FMRP is elevated in breast tumors and its high expression correlates with lung and LN metastasis [144].

No information with respect to *ABHD11-AS1* in BC was found in PubMed; however, *ABHD11-AS1* is increased in colorectal carcinoma (CRC) [145], endometrial carcinoma [146], ovarian cancer [147], papillary thyroid cancer [148], and pancreatic cancer [149], implicating an oncogenic role for *ABHD11-AS1* in these cancers. *ABHD11-AS1* was identified in network 2 (Appendix A, Figure 7) and was depicted as negatively regulating miR-1254, which targets *RBBP6* (RB binding protein 6, ubiquitin ligase). KM plotter [150] revealed no significant difference in OS in BC patients related to *ABHD11-AS1* expression (data not shown).

Although PubMed contained no reports on *NALT1* in BC, *NALT1* was overexpressed in gastric cancer (GC), associated with reduced OS, and was found to promote the invasion of the normal human gastric epithelial GES-1 cell line and GC cancer cell lines in vitro by suppressing NOTCH signaling [151]. Further studies are needed to determine the role and expression of NALT1 in BC.

*TINCR* was reported to be overexpressed and oncogenic in HER2+ breast tumors [69]. Higher *TINCR* expression in breast tumors (all types) was associated with reduced OS [69]. This study showed that *TINCR* expression was higher in MDA-MB-453 HER2+ BC cells compared to UACC-812 TNBC, BT549 TNBC, MDA-MB-231 TNBC, and MCF-7 Luminal A BC cells. TINCR acted as a ceRNA for miR-125b, relieving the repression of *ERBB2* in UACC-812 cells [69]. *TINCR* was identified in network 1 (Appendix A, Figure 6) and was depicted as positively regulating KRT78 (keratin 78, gene *KRT14*). *TINCR* was reported to interact directly with mRNAs in human epidermal differentiation and barrier formation [152]. KRT14-expressing BC cells are invasive and metastatic, forming clusters for dissemination and colonization in metastatic niches [153,154]. The increase in *TINCR* in response to miR-29b-1/a transfection appears to oppose their anti-tumorigenic activity in BC cells.

The *MINCR* level was higher in MCF-7 cells than in LCC9 cells and was decreased by anti-miR-29 in MCF-7 cells transfected with either pre-miR-29a-3p or pre-miR-29b-1-3p, suggesting the upregulation of *MINCR* by miR-29b-1/a in these cells (Table 2). No information with respect to *MINCR* in BC was found in PubMed; however, *MINCR* was upregulated in CRC tumors [155] and in non-small cell lung cancer (NSCLC) [156]. KM plotter [150] revealed no significant difference in OS in BC patients related to *MINCR* expression (data not shown).

Mutations were identified in the promoter of *NEAT1* that increased its expression in BC [157]. *NEAT1* was overexpressed in luminal A, luminal B, HER2+, and basal-like (TNBC) tumors [158]. Patients whose primary breast tumors showed high expression of *NEAT1* had shorter OS [138]. *NEAT1* was elevated in the plasma of BC patients and associated with LN positivity and TNBC tumor type [159]. *NEAT1* is involved in the organization of nuclear paraspeckles for gene transcription and splicing [42]. Nuclear speckles are dynamic punctate compartments in the nucleus that contain components of the pre-mRNA spliceosome, including serine/arginine-rich splicing factors (SRSFs), small nuclear ribonucleoproteins (snRNPs), RNA polymerase (Pol) II subunits, 3′ end processing proteins, m6A writers METTL3/METTL14, m6A reader YTHDC1, and various protein kinases that regulate the pool of proteins in the speckles [160,161]. *NEAT1* was identified as an essential component of the FOXN3-SIN3A repressor complex and overexpression of *NEAT1* promoted EMT in MCF-7 cells and lung metastasis of MCF-7 cells when orthotopically implanted in the mammary fat pad of immunocompromised female mice, suggesting that *NEAT1* has oncogenic and pro-metastatic activity [162]. *NEAT1* was also identified in a gene (*ESR1, DKC1*)–lncRNA (*TERC* and *TUG1*) interaction network in breast tumors from The Cancer Genome Atlas (TCGA) [163]. Increased *NEAT1* was detected in cisplatin- and taxol- resistant MDA-MB-231 cell lines compared to parental MDA-MB-231 cells, and knockdown of *NEAT1* inhibited MDA-MB-231 xenograft tumor growth in vivo [164]. *NEAT1* has been shown to be a ceRNA for a number of miRNAs, including miR-124, thus upregulating *STAT3* [165]; for miR-133b, thus de-repressing *TIMM17A* [166]; for miR-141-3p, thus increasing *KLF12* [167] in MCF-7 and MDA-MB-231 cells; for miR-107, thus upregulating *CPT1A* in HEK-293 cells [168]; and for miR-205-5p, thus de-repressing VEGFA in CRC cells [169]. *NEAT1* was identified in network 1 (Appendix A, Figure 6) and was depicted as negatively regulating miR-1321, miR-361-5p, and miR-1246. miR-361-3p was upregulated in fulvestrant-resistant MCF-7 cells [170] and targets *GLI1* (GLI family zinc finger 1, a transcription factor), which is increased in breast tumors and inversely correlates with disease-free survival (DFS) in luminal A tumors (reviewed in [22]). *NEAT1* was identified in network 3 (Appendix A, Figure 8), in which it was depicted as negatively regulating miR-185-5p and miR-101-3p. miR-185-5p expression is reduced in breast tumors and miR-185-5p targets *VEGFA, E2F6*, and *DNMT1* (reviewed in [129]). miR-101 is a tumor suppressor that targets *ZEB1* and *ZEB2* (reviewed in [22]). Overall, the increase in *NEAT1* by miR-29b-1/a in MCF-7 and LCC9 cells seems to oppose the mechanisms of anti-tumor activity of these miRNAs.

*MALAT1* was ~5.5-fold higher in MCF-7 cells than in LCC9 cells and is an established miR-29 interactor (Table 2). *MALAT1* is a well-studied lncRNA (1573 papers in PubMed) that is evolutionarily conserved and highly expressed across all tissues (reviewed in [171]). *MALAT1* was originally identified as an oncogene in non-small cell lung cancer [172]. *MALAT1* is upregulated in multiple myeloma and in many solid tumors, including breast tumors (reviewed in [173,174]). *MALAT1* is oncogenic and promotes tumor progression and metastasis in various cancers, including BC (reviewed in [21,175].) Patients whose primary breast tumors showed high expression of MALAT1 had shorter OS [138]. In addition to tumor expression, one study reported higher serum levels of *MALAT1* in BC patients (n = 157) compared to control women (n = 107) [176]. *MALAT1* expression is associated with ERα+/PR+ breast tumors and with lower relapse-free survival (RFS) [138]. However, *MALAT1* expression was associated with decreased DFS in patients with HER2+ and TNBC tumors [177]. *MALAT1* increases with breast tumor stage and was 2–3 times higher in lung and brain metastases when compared to matched primary luminal breast tumor sections [175]. Other studies have reported higher *MALAT1* levels in breast tumors than in normal breast tissue [178]. *MALAT1* is oncogenic in BC and upregulates the WNT/β-catenin (*CTNNB1*) pathway [179]. *MALAT1* mutations are frequent in breast tumors [180,181]. A meta-analysis showed that high *MALAT1* expression is associated with reduced OS and RFS in BC patients [182].

*MALAT1* is a nuclear-localized lncRNA that acts as a scaffold to position nuclear speckles at active gene loci [183]. *MALAT1* is m6A-modified and interacts with m6A reader YTHDC1 in esophageal cancer cells [184]. Although *MALAT1* is m6A-modified, the question of how this affects its activities in BC cells remains uncertain. A network analysis of ncRNAs in cancer drug resistance-associated lncRNAs/miRNAs and TAM resistance identified a ‘hub’ with lncRNAs *MALAT1* and *CCAT2*; miR-29a/b, miR-148, miR-152, miR-206, miR-221, miR-222, miR-335, miR-375, miR-26a/b, and miR-27b [47]. *MALAT1* acts as a ceRNA for miR-9, miR-26a/b, miR-101b, miR-133, miR-145-5p, miR-195, miR-200s, miR-205, miR-206, miR-376a, and miR-503 (reviewed in [171,173,174]). Proteins interacting with *MALAT1* in a whole-cell lysate of HEPG2 human HCC cells included proteins involved in RNA processing, splicing, and gene transcription [185]. *MALAT1* is a therapeutic target using antisense oligonucleotides (ASO) and ‘gapmers’ with a central DNA flanked by modified oligonucleotides that interact with *MALAT1* and degrade it via nuclear RNase H [186,187]. *MALAT1* was identified in networks 1 and 3 (Appendix A, Figure 6 and Figure 7) and negatively regulates *SMAD2* (SMAD family member 2) and miR-185-5p, discussed previously as being reduced in breast tumors (reviewed in [129]. Overall, the increase in *MALAT1* by miR-29b-1/a in MCF-7 and LCC9 cells appears to oppose their anti-cancer activities.

### 3.4. lncRNAs Differentially Expressed in Endocrine-Sensitive MCF-7 versus Endocrine-Resistant LCC9 Cells and Their Roles in BC

We identified 53 lncRNAs differentially expressed in MCF-7 and LCC9 cells that were not regulated by miR-29b-1/a. Of these, 17 had low expression levels in MCF-7 cells, i.e., FPKM ≤ 1, and ten had no published role in BC tumors or cell lines (Appendix A). The roles of seven lncRNAs (P*CAT1, CAHM, HOXA-AS2, MIR2052HG, BDNF-AS, CASC15*, and *HOXA11*-AS) in BC are summarized in Appendix A. Thirty-five lncRNAs were expressed at FPKM ≥ 1 and their roles in BC are reviewed below. Of these, twenty-seven showed higher abundance in MCF-7 than LCC9 cells: *PCGEM1, KRT7-AS1, SATB2*-*AS1, HAGLR, HAR1B, VLDLR-AS1, ZEB1-AS1, FTX, CDKN2B-AS1, PCAT6, HOTAIRM1, MIR22HG, LINC-PINT, NBR2,TMEM161B-AS1, HAR1A, MIR503HG, PSMD6-AS1, DHRS4-AS1, MIR600HG, NORAD, XIST, PVT1, SNHG1*, and *ZFAS1* (Table 3). The lncRNAs *FOXP4-AS1, H19, HMMR-AS1, FOXD3-AS1, PPP1R12A-AS1, LINC01116, HOTAIR, DLEU2, MIF-AS1*, and *TP53TG1* were more abundant in LCC9 cells than in MCF-7 cells. GO processes and network analysis for these differentially expressed lncRNAs are summarized in Appendix A, respectively. Selected networks are shown in Figure 9, Figure 10, Figure 11 and Figure 12 and Appendix A. Only one lncRNA, *APOBEC3B-AS1*, was commonly downregulated in LCC9 vs. MCF-7 cells in both our analysis and in the Agilent human lncRNA + mRNA Array V4.0 reported previously [49]. However, the avg. FPKM was 1.45 in MCF-7 and 0.033 in LCC9 cells, suggesting low abundance. There are no reports about *APOBEC3B-AS1* in PubMed.

### 3.5. lncRNAs More Highly Expressed in Endocrine-Sensitive MCF-7 versus Endocrine-Resistant LCC9 Cells and Their Roles in BC

*PCGEM1* (LINC00071) is a scaffolding lncRNA that plays a role in the transcription of androgen receptor (AR) target genes in prostate cancer (PCa) cell lines (reviewed in [188]). *PCGEM1* was not identified in LCC9 cells and showed low expression in MCF-7 cells (Table 3). *PCGEM1* was identified in networks 1 and 3 (Figure 9 and Figure 10). *PCGEM1* was reported to physically associate to a subset of the metabolic gene promoters (*CANT1, CYP11A1, DHCR24, FASN, G6PD* (shown in Figure 10), *GLS, GPI, GSR, HK2, IDH1, IDH2,* and *LDHA*) [189].

*KRT7-AS* is m6A modified by METTL3 and forms an RNA hybrid with *KRT7* to stabilize that transcript, and *KRT7-AS* promotes lung metastasis from MDA-MB-231 and BT-549 cells [190]. There are no known functions of *SATB2-AS1* in BC. A recent report showed that *HAGLR* overlaps with miR-7704, which represses *HAGLR* expression in MCF-7, MDA-MB-231, and MCF-10A cells [191]. High *HAGLR* expression was associated with lower RFS in BC patients [192].

No reports of *HAR1B* or *VLDLR-AS1* in BC were found. Decreased *HAR1B* expression levels are associated with poor prognosis in HCC [193].

*ZEB1-AS1* is a well-recognized cancer-related lncRNA that has been identified as an oncogene in diverse malignancies [194]. It is associated with several functional roles, including EMT, proliferation, migration, invasion, and metastasis by regulating multiple genes including miR-200s [195]. *ZEB1-AS1* was upregulated in TNBC cell lines and tumors and stabilized *ZEB1* mRNA by binding with ELAVL1 (*ELAV1*, ELAV-like RNA binding protein 1), forming a feedback loop to promote TNBC progression [196]. *ZEB1-AS1* interacts with miR-505-3p (Figure 11). The opposite strand miRNA-miR-505-5p was downregulated in the serum of BC patients compared to that of healthy controls [197]. ZEB1-AS1 is a ceRNA for miR-505-3p, thus de-repressing *TRIB2* (tribbles pseudokinase 2) in pancreatic cancer [198].

*FTX*, a chromatin-associated lncRNA, is regulated by pathways mediating the initiation and progression of breast tumors [199]. The *FTX* gene harbors two miRNA clusters: miR-374b/421 and the miR-545/374a cluster, which were upregulated in HCC tissues and associated with a poor prognosis [200]. Estrogen-related receptor gamma (ERRγ, *ESRRG*) was reported as a target of miR-545 [200]. ERRγ is upregulated in 75% of breast tumors [201] and induces TAM resistance [202]. *ESRRG* transcript levels were low in both MCF-7 and LCC9 cells (FPKM < 1, data not shown).

*CDKN2B-AS1* (ANRIL) is located in the 9q21.3 region with rs62560775, associated with lung adenocarcinoma and BC susceptibility [203]. *CDKN2B-AS1* is upregulated in MCF10A breast epithelial cells [204] and in breast tumors [205]. *CDKN2B-AS1* was identified as a member of the ceRNA network for MMP1/MMP11-miR-145-5p and speculated to be involved in the development of early BC [205]. Network analysis showed *CDKN2B-AS1* stimulating HDAC (histone deacetylases) (Appendix A).

*PCAT6* is upregulated in TNBC tissues and cells [206]. *PCAT6* acts as a sponge for miR-4723-5p to upregulate *KDR* (VEGFR2, vascular endothelial growth factor receptor 2) [206]. Knockdown of *PCAT6* promotes the radiosensitivity of MDA-MB-468 and MDA-MB-231 cells by inhibiting proliferation and inducing apoptosis [207]. This occurs with *PCAT6* directly targeting and negatively regulating the expression of miR-185-5p to modulate *TPD52* (tumor protein D52) expression [207]. Network analysis showed E2F1 simulating *PCAT6* (Appendix A).

*HOTAIRM1* was reported to be overexpressed in basal-like breast tumors [90] and showed higher expression in TAM-resistant MCF-7 cells [208]. These results are in contrast with our observation of ~4-fold higher expression of *HOTAIRM1* in MCF-7 cells compared to LCC9 cells.

We previously reported that the expression of *MIR22HG* was downregulated in response to the treatment of MCF-7 BC cells with the anti-cancer phenolic lipid anacardic acid T [209]. *MIR22HG* is a tumor suppressor and high *MIR22HG* was associated with increased OS in BC samples in an analysis of data from TCGA database [210]. *MIR22HG* was suggested to be a ceRNA for miR-424 [211]. Network analysis showed ERRα regulating *MIR22HG* and *LINC-PINT* (Figure 9). In a pan-cancer dataset of 15 BC tissues, *LINC-PINT* expression was downregulated compared to normal tissue [212]. High expression of *LINC-PINT* was associated with favorable DFS in BC patients [212].

The lncRNA *NBR2* is encoded ‘head-to-head’ with tumor suppressor *BRCA1* [213]. The expression of *NBR2* and *BRCA1* are affected by the SNP rs9911630 [214]. Upon energy stress, i.e., glucose deprivation, *NBR2* expression was increased in MDA-MB-231 TNBC cells and in other cancer cells [215]. Network analysis showed ERRα regulating *NBR2* (Figure 9). *NBR2* was shown to interact with AMP-activated protein kinase (AMPK; a critical sensor of cellular energy status) to potentiate the AMPK kinase activity and increase G*LUT1* expression [215]. *NBR2* expression, which has been associated with higher OS in BC and *NBR2*, acts like a tumor suppressor in MDA-MB-231 xenograft tumors in vivo [216]. It is yet to be determined whether *NDR2* regulates BRCA1 but the >2-fold higher expression of *NBR2* in MCF-7 cells compared to LCC9 cells suggests an altered metabolism in LCC9 cells, as reported previously [217,218,219].

In combination with four other lncRNAs, higher *TMEM161B-AS1* was reported to be a predictor for tumor recurrence in BC patients [192]. An in silico analysis using the HGNC (HUGO Gene Nomenclature Committee) database (Bethesda, MD, USA) predicted *TMEM161B-AS1* to be associated with miR-17-5p and *MAPK14* [220].

*HAR1A* was among a signature of nine other lncRNAs identified in TCGA of which the upregulation predicted recurrence in invasive BC [221]. Network analysis showed ERRα regulating *HAR1A* (Figure 9). No reports on ERRα regulating HAR1A in BC were found in PubMed.

Low *MIR503HG* expression was detected in TNBC tissues, and in MDA-MB-231 and TB549 TNBC tissues, in which MIR503HG serves as a tumor suppressor [222,223]. Low MIR503HG expression was associated with a worse prognosis and was correlated with clinical stage, LN metastasis, and distant metastasis in TNBC patients. In vitro upregulation of *MIR503HG* inhibited MDA-MB-231 and MDA-MB-453 TNBC cell migration and invasion. Two pathways—the miR-103/*OLFM4* axis and the miR-224-5p/HOXA9 axis [222]—have been implicated in mediating the functions of *MIR503HG* [223]. Network analysis showed ERRα to regulate *MIR503HG* (Figure 9). The lower abundance of *MIR503HG* in LCC9 cells is in agreement with their higher proliferative rate and greater invasion and migration abilities compared to MCF-7 cells [224].

Initially identified in an lncRNA microarray study, *PSMD6-AS1* expression levels were significantly higher in ER/PR(+) versus ER/PR(-) BC patients and in postmenopausal versus premenopausal BC patients [225]; however, the functional role of *PSMD6-AS1* in ER + BC is yet to be determined.

*DHRS4-AS1* and *MIR600HG* are tumor suppressors in human cancer and their higher expression in MCF-7 cells compared to LCC9 cells fits the endocrine-resistant phenotype of these cells. *DHRS4-AS1* was downregulated in NSCLC and mediated its effects through a TP53- and TET1-associated *DHRS4-AS1*/miR-224-3p axis [226]. *MIR600HG* is downregulated in CRC and its expression has been inversely correlated with OS [227].

*NORAD* is oncogenic and increased in BC tissues, MCF-7, and MDA-MB-231 cells, and is correlated with reduced OS [228]. *NORAD* knockdown reduced proliferation, invasion, and migration of MCF-7 and MDA-MB-231 BC cells and reduced tumor growth in vivo [228]. *NORAD* stimulated TGF-β signaling and directly increased *RUNX2* expression, resulting in BC progression and metastasis [228]. The *NORAD* level was higher in luminal A tumors compared to basal-like or TNBC breast tumors [229]. High expression of NORAD in basal-like cancers was associated with lower OS; however, *NORAD* offered no prognostic information in luminal A BC tumors [229].

*XIST*, which is involved in X-inactivation and genomic imprinting, has a tumor suppressive role in BC [89,230]. The abundance of *XIST* was negligible in LCC9 cells and high in MCF-7 cells (Table 3). Network analysis depicted *XIST* as downregulating miR-140-5p and interacting with miR-20a-5p (Figure 10 and Figure 11). *XIST* expression was low in primary breast tumors and their metastasis [89]. Ectopic expression of *XIST* in MCF-7 cells reduced AKT phosphorylation and cell viability—a process that was shown to be under epigenetic regulation via the recruitment of HDAC3 to the *PHLPP1* promoter [89]. The tumor suppressive role of *XIST* in BC occurs in part through the miR-155/*CDX1* axis [230]. In contrast with these reports, other studies have reported XIST to be higher in breast tumors than in normal breast tissue [231]. *XIST*, a direct target of miR-7, was inversely associated with miR-7 in breast tumors [231]. Ectopic expression of miR-7 was shown to bind directly to *XIST* and reduce its expression and to reduce BC stem cell-driven tumor growth in vivo [231]. Consistently with its tumor suppressor role, the knockdown of XIST increased M1-to-M2 macrophage phenotype polarization and promoted the cell proliferation and migration of breast and ovarian cancer cells by competing with miR-101 and inhibiting C/EBPα and KLF6 expression [232].

*PVT1* expression is upregulated in breast tumors and cell lines and associated with BC risk [90,233,234]. Serum levels were also higher in BC patients [5]. *PVT1* expression is positively correlated with miR-1207-5p (a *PVT1*-derived miRNA) and the estrogen-treatment-induced expression of *PVT1* and miR-1207-5p in T47D BC cells [235]. *PVT1* expression was negatively correlated with the pathological stage and the levels of ER, HER2, and p53, and was positively correlated with PR in multiple primary neoplastic tissues [236]. Network analysis indicated that *PVT1* downregulates miR-186-5p (Figure 10). Knockdown of *PVT1* inhibited growth and motility, and induced apoptosis in MCF-7 and MDA-MB-436 BC cells [237]. In vivo, knockdown of *PVT1* reduced tumor volume and weight [237]. Multiple mechanisms have been described in order to understand *PVT1*’s role in breast tumorigenesis. *PVT1* suppression enhanced *TRPS1* levels by negatively targeting miR-543 in BC [238]. *PVT1* binds KLF5, an interaction that is enhanced by BAP1 (BRCA1-associated protein), to upregulate beta-catenin signaling and promote TNBC tumorigenesis [237]. *PVT1* is a ceRNA for miR-186 in multiple cancers [239]. For example, in gastric cancer, *PVT1*–miR-186 interaction inhibits HIF-1α expression and promotes cell proliferation and invasion [240].

*SNHG1* is associated with endocrine cancers, including BC [241]. *SNHG1* is upregulated in breast tumors and cell lines and promotes cell migration, invasion, and proliferation in vitro, as well as MDA-MB-231 ‘metastasis’ and colonization in the lungs of immune-compromised female mice after tail-vein injection [242]. S*NHG1* is a ceRNA and reduces miR-382-5p [243], miR-193a-5p [242], and miR-573 [244] in BC cells. Knockdown of *SNHG1* reduced BC cell proliferation, migration, invasion colony formation, and EMT [243]. *SNHG1* inhibited the differentiation of regulatory T cells (Tregs), reduced miR-448 expression, and increased *IDO1* (indoleamine 2,3-dioxygenase 1), which is implicated in the immune escape of BC cells [245]. *SNHG* is a ceRNA for miR-140-5p in cholangiocarcinoma [246] and glioma [247]. Whether this interaction occurs in BC cells is unknown.

*ZFAS1* is a host to three C/D box snoRNAs, which target rRNAs for post-transcriptional modification [248]. *ZFAS1* was identified among the highest and most differentially expressed transcripts during mouse mammary gland development, i.e., decreasing 10-fold between pregnancy and lactation [249]. Network analysis showed *ZFAS1* downregulating miR-186-5p and interacting with *ESR1* (Figure 10 and Figure 11). The abundance of *ZFAS1* is reduced in breast tumors compared to normal breast tissue [249] and in BC cell lines compared to MCF-10A cells [250]. *ZFAS1* ectopic expression significantly suppressed cell proliferation by causing cell cycle arrest and inducing apoptosis in MCF-7 and MDA-MB-436 BC cells [250]. *ZFAS1* was predominantly associated with the 40S small ribosomal subunit in MDA-MB-468 TNBC cells [248]. The lower expression of *ZFAS1* in LCC9 cells compared to MCF-7 cells fits with the tumor-suppressive activity of this lncRNA.

### 3.6. lncRNAs More Highly Expressed in Endocrine-Resistant LCC9 Cells than in MCF-7 Cells and Their Roles in BC

We observed that *FOXP4-AS1* was more abundant in LCC9 cells than in MCF-7 cells (Table 3). *FOX4P* is also higher in LCC9 cells than in MCF-7 cells (data not shown). There are no reports of *FOXP4-AS1* in BC; however, *FOXP4-AS1* expression is high in prostate tumors and PCa cell lines [251]. *FOXP4-AS1* acts as a ceRNA for miR-3184-5p and increases *FOXP4* expression, acting in an oncogenic pathway in prostate cancer [251]. *FOXP4-AS1* is also upregulated and oncogenic in HCC [252] and gastric cancer [253]. In contrast, *FOXP4-AS1* upregulation in ovarian cancer is associated with higher OS, suggesting cell-type-specific regulatory roles of *FOXP4-AS1* in different tumors [254].

*H19* is increased and plays an oncogenic role in a variety of cancers, including BC, where it acts as a ceRNA for various miRNAs (reviewed in [255]). Fulvestrant increased *H19* expression and *H19* was higher in LCC2 and LCC9 cells than in MCF-7 cells [256]. *H19* targets miR-29b-3p in rat cardiomyocytes [257], bladder cancer cells [258], and CRC cells [259].

*HMMR-AS1* was higher in breast tumors than in normal breast tissue, with the highest levels found in basal-like tumors, followed by HER2+ tumors, which showed higher expression than luminal A or B breast tumors [260]. Knockdown of *HMMR-AS1* inhibited the proliferation, migration, and invasion of MDA-MB-231 and MDA-MB-468 basal-like TNBC cell lines [260].

*FOXD3-AS1* [261] and *LINC01116* [262] were more highly expressed in breast tumors compared to normal breast tissue. High expression *LINC01116* correlated with reduced OS, tumor size, and TNM stage. *LINC01116* is a ceRNA for miR-145, resulting in increased ERα.

*HOTAIR* abundance was ~3 times higher in LCC9 cells than in MCF-7 cells (Table 3). *HOTAIR* was one of the first characterized lncRNAs with a conserved structure that interacts with over 70 proteins (reviewed in [129]). *HOTAIR* acts as a nuclear scaffold for the PRC2 and LSD1 histone modifying complexes to promote histone H3K27 methylation and H3K4 demethylation to silence target genes and promote BC metastasis [48]. *HOTAIR* is upregulated in breast tumors and is a ceRNA for miR-20a-5p [263]. In agreement with our data (Table 3), *HOTAIR* is upregulated in endocrine-resistant BC cells and its overexpression activates ERα transcriptional activity independently of ligands [264]. *HOTAIR* is also increased in TAM-resistant human breast tumors [264]. High expression of *HOTAIR* in exosomes in serum from BC patients was associated with lower RFS and OS [265]. ERα interacts directly with *HOTAIR* in MCF-7 cells [266]. Overexpression of *HOTAIR* in MCF-7 cells grown under hormone-free (serum-starved) medium conditions increases the number of DNA sites to which ERα binds in chromatin immunoprecipitation (ChIP) assays and increases the mRNA expression of some ERα target genes, e.g., *GREB1, TFF1, PGR,* and *CTSD* [264]. This is depicted in network 3: *PCGEM1, ZFAS1 RNA, ZEB1-AS1, HOTAIR,* and *ESR1*, shown in Figure 11.

*DLEU2* is an oncogene in multiple malignancies [267,268,269]. D*LEU2* expression was increased by MVLN (MCF-7-derived) BC cells and abolished by 4-hydroxytamoxifen (4-OHT, an active TAM metabolite) in a process that was independent of protein synthesis [270]. Tumor suppressor miRs miR-15a and miR-16 are transcribed from the *DLEU2* locus [271]. Network analysis showed *DLEU2* downregulating miR-186-5p (Figure 12).

*MIF-AS1* abundance was approximately nine times higher in in LCC9 cells than in MCF-7 cells (Table 3). *MIF-AS1* was upregulated in BC tissues and cells, including MCF-7, MDA-MB-231, and MDA-MB-468 cells [272]. Low *MIF-AS1* expression was associated with poor OS. The repression of *MIF-AS1* inhibited cell proliferation, migration, and EMT markers in MCF-7 and MDA-MB-231 BC cells [272]. By functioning as a ceRNA, *MIF-AS1* modulated the miR-1249-3p/HOXB8 axis, resulting in increased *HOXB8* (Homeobox B8) expression [272].

*TP53TG1* (*LINC00096*) abundance was approximately three times higher in LCC9 cells than in MCF-7 cells (Table 3). *LINC00096* was identified in a microarray screening study to be the most significantly increased LncRNA in TNBC tissues and cells [273]. Loss-of-function assays indicated that *LINC00096* suppression inhibited cell proliferation and invasion through regulation of the miR-383-5p/*RBM3* (RNA binding motif protein 3) pathway in BT-549 and MDA-MB-231 cells [273]. Other studies have reported *TP53TG1* to be a tumor suppressor in CRC, due to epigenetic inactivation [273]. *TG53TG1* expression is stimulated by DNA damage and depends on a wild-type TP53 expression in breast tumors [274]. Network analysis showed that *TG53TG1* is regulated by ERRα (Figure 9).

Taken together, the lncRNAs differentially expressed in MCF-7 endocrine-sensitive and LCC9-endocrine-resistant breast cancer implicate a network of miRNAs and genes in pathways known to regulate cell proliferation, invasion, and cell signaling in breast cancer.

## 4. Conclusions

This is the first examination of the impact of modulating the expression of miR-29b-1-3p and miR-29a-3p on lncRNA abundance in TAM- and fulvestrant-sensitive (MCF-7) versus resistant (LCC9) ER+ BC cells. Some of the miR-29b-1/a–lncRNA interactions identified here appear to be direct interactions, as indicated in the DIANA-LncBase v.3 database; however, a number of new potential interactions were detected that require further confirmation. In addition to the miR-29b-1-3p- and miR-29a-3p-regulated lncRNAs, we also identified cell-line-specific differences in lncRNA expression in MCF-7-endocrine-sensitive and LCC9-endocrine-resistant BC cells. The networks and GO processes identified in the analysis of these lncRNAs provide new insights into the contributions of lncRNAs to endocrine resistance. Further experiments are needed in order to elucidate these mechanisms in endocrine-resistant ER+ BC in vivo.

## Figures and Tables

**Figure 1 cancers-13-03530-f001:**
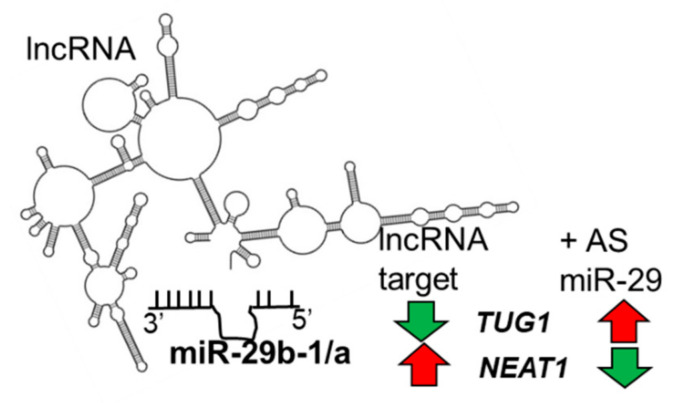
Model of identification of miR-29b-1/a regulation of lncRNAs in MCF-7 and LCC9 breast cancer cells. If an lncRNA (drawn in BioRender.com) is a target of miR-29b-1/a, then antisense (AS) miR-29 will block the effect of miR-29b-1/a on that lncRNA. The green arrow represents a decrease and the red arrow represents an increase in the lncRNA abundance in response to miR-29b-1-3p and or miR-29a-3p transfection that is blocked by AS-miR-29 (arrows in opposite directions). An example of a downregulated and an upregulated lncRNA detected in both MCF-7 and LCC9 cells from Table 1 and Table 2 is shown.

**Figure 2 cancers-13-03530-f002:**
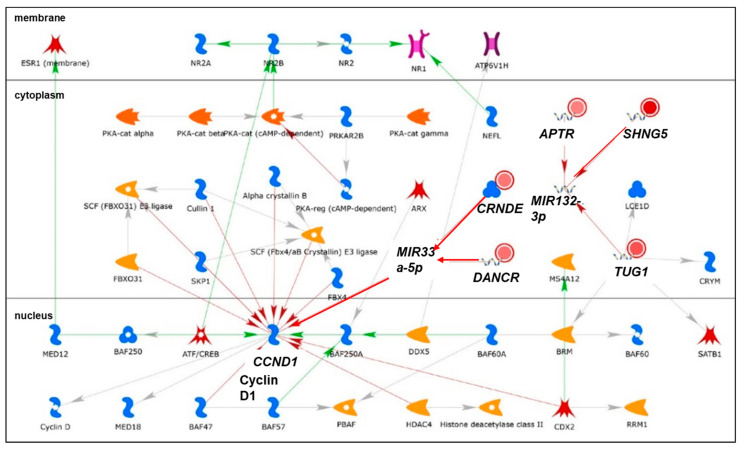
Network 2: *TUG1, CRNDE, APTR, DANCR,* and *SNHG5*, identified in lncRNAs downregulated by miR-29b-1-3p and miR-29a-3p in MCF-7 and/or LCC9 cells by MetaCore analysis. Green lines with arrows = stimulation; red lines with arrows = inhibition.

**Figure 3 cancers-13-03530-f003:**
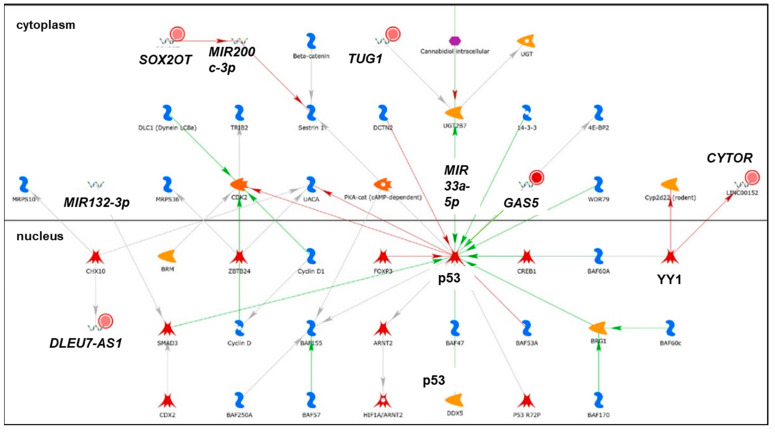
Network 3: *DLEU7-AS1, GAS5, SOX2OT, LINC00152,* and *TUG1*, identified in lncRNAs downregulated by miR-29b-1-3p and miR-29a-3p in MCF-7 and or LCC9 cells by MetaCore analysis. Green lines with arrows = stimulation; red lines with arrows = inhibition.

**Figure 4 cancers-13-03530-f004:**
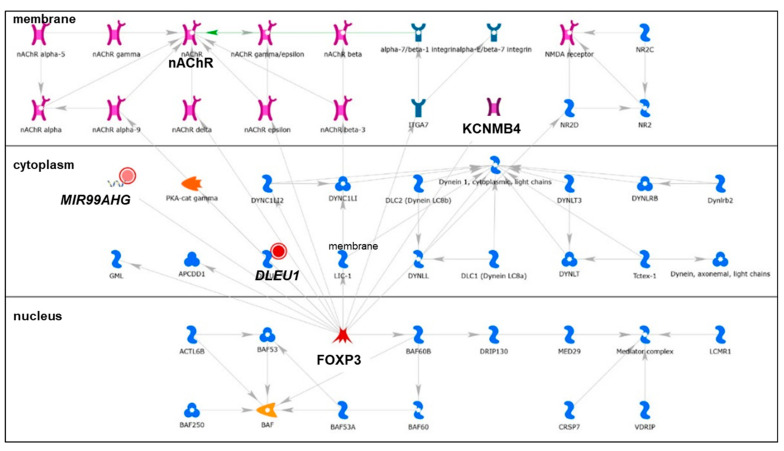
Network 4: *DLEU1, MIR99AHG, FOXP3, KCNMB4*, and nAChR delta, identified in lncRNAs downregulated by miR-29b-1-3p and miR-29a-3p in MCF-7 and/or LCC9 cells by MetaCore analysis. Green lines with arrows = stimulation; red lines with arrows = inhibition. nAChR = nicotinic acetylcholine receptor delta (*CHRND*). nAChRs are present in breast tumors (reviewed in [65]).

**Figure 5 cancers-13-03530-f005:**
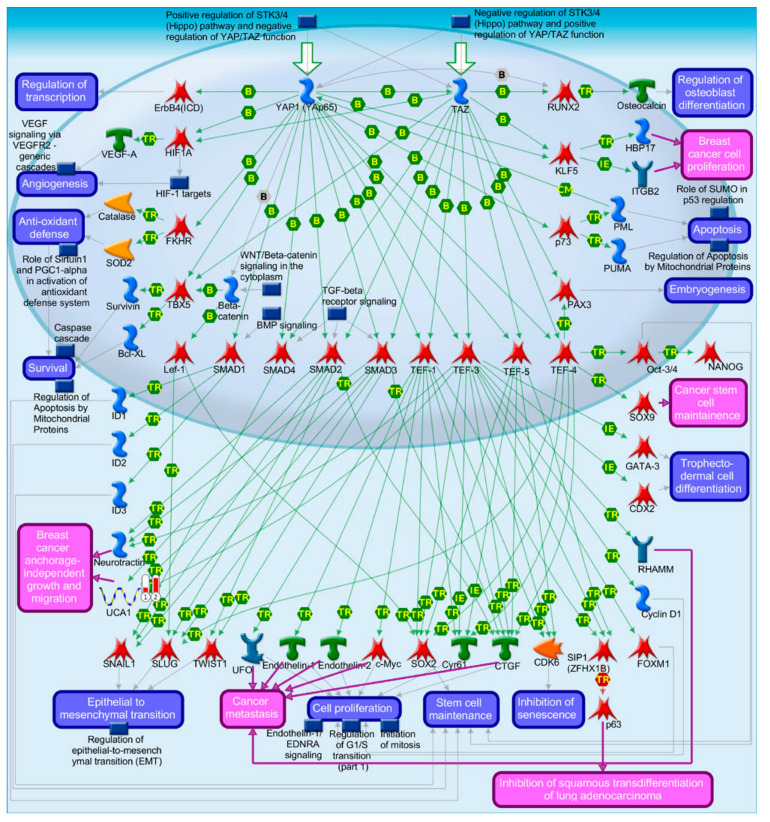
Pathway Map: development: YAP/TAZ-mediated co-regulation of transcription. The lncRNA UCA1 was upregulated by miR-29b-1/a in MCF-7 and LCC9 cells. Image is from MetaCore analysis.

**Figure 6 cancers-13-03530-f006:**
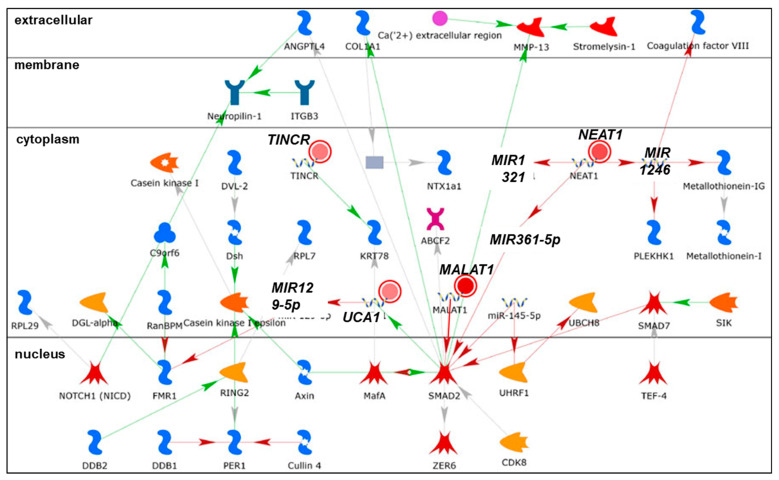
Network 1: NEAT1, UCA1, MALAT1, TINCR, and SMAD2, identified in lncRNAs upregulated by miR-29b-1-3p and miR-29a-3p in MCF-7 and/or LCC9 cells by MetaCore analysis; green lines with arrows = stimulation; red lines with arrows = inhibition.

**Figure 7 cancers-13-03530-f007:**
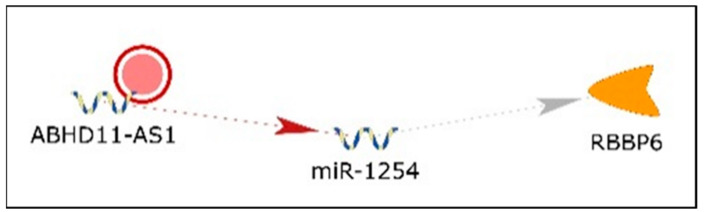
Network 2: *ABHD11-AS1*, miR-1254, and *RBBP6*, identified in lncRNAs upregulated by miR-29b-1-3p and miR-29a-3p in MCF-7 and/or LCC9 cells by MetaCore analysis. Red line with arrow = inhibition. The dotted gray line is a putative regulation.

**Figure 8 cancers-13-03530-f008:**
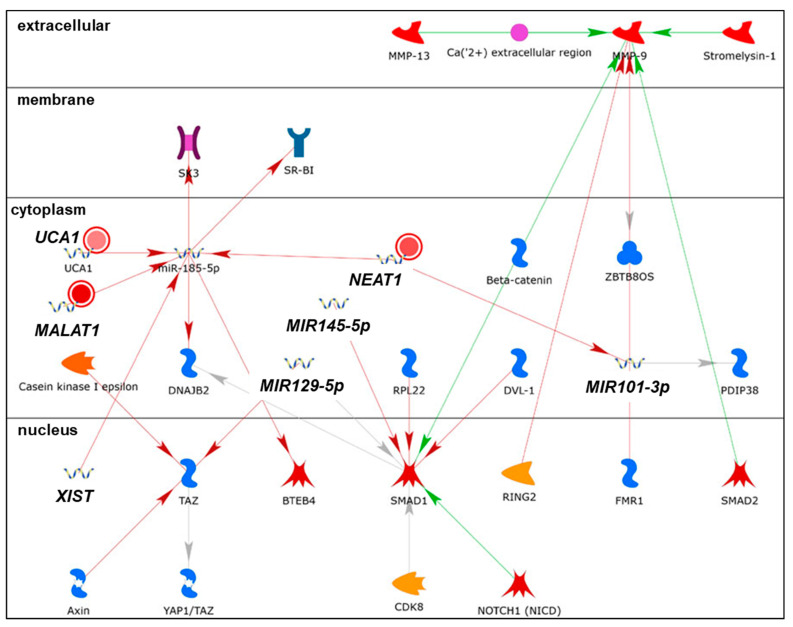
Network 3: *NEAT1, MALAT1, UCA1*, miR-185-5p, and S*MAD1*, identified in lncRNAs upregulated by miR-29b-1-3p and miR-29a-3p in MCF-7 and/or LCC9 cells by MetaCore analysis. Green lines with arrows = stimulation; red lines with arrows = inhibition.

**Figure 9 cancers-13-03530-f009:**
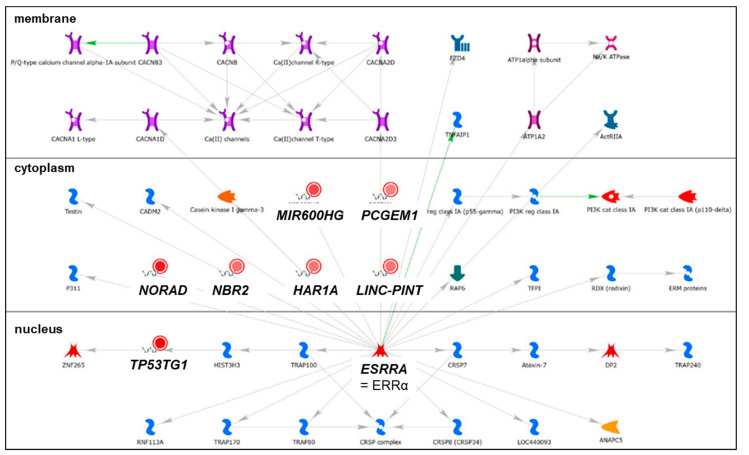
Network 1: LOC647979 (*NORAD*), *NBR2, PCGEM1, LINC-PINT,* and *MIR600HG*, identified in lncRNAs differentially expressed in MCF-7 and or LCC9 cells by MetaCore analysis. Green lines with arrows = stimulation; red lines with arrows = inhibition.

**Figure 10 cancers-13-03530-f010:**
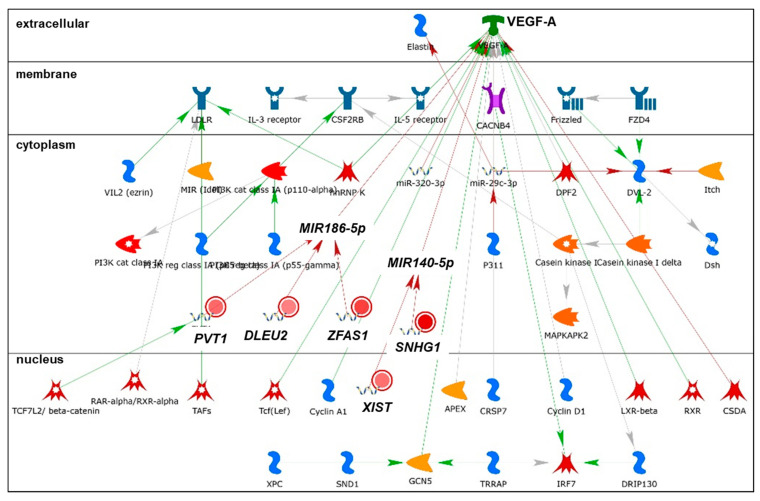
Network 2: *PVT1, ZFAS1 RNA, XIST, SNHG1,* and *DLEU*, identified in lncRNAs differentially expressed in MCF-7 and or LCC9 cells by MetaCore analysis. Green lines with arrows = stimulation; red lines with arrows = inhibition.

**Figure 11 cancers-13-03530-f011:**
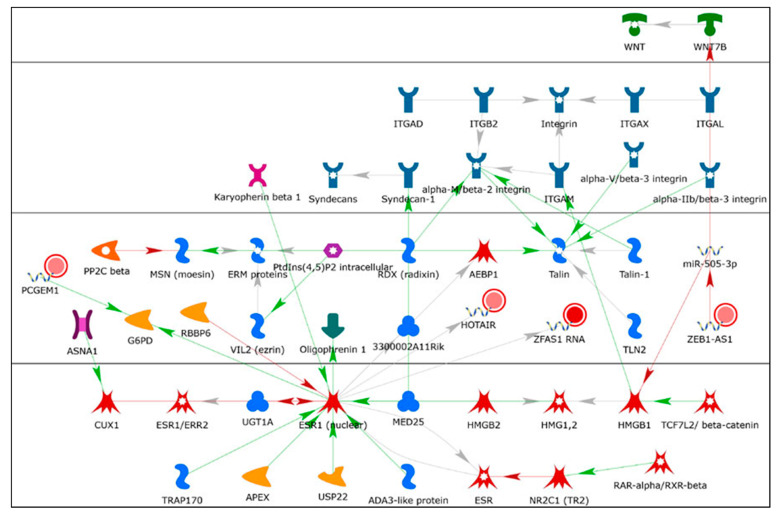
Network 3: *PCGEM1, ZFAS1 RNA, ZEB1-AS1, HOTAIR,* and *ESR1*, identified in lncRNAs differentially expressed in MCF-7 and or LCC9 cells by MetaCore analysis. Green lines with arrows = stimulation; red lines with arrows = inhibition.

**Figure 12 cancers-13-03530-f012:**
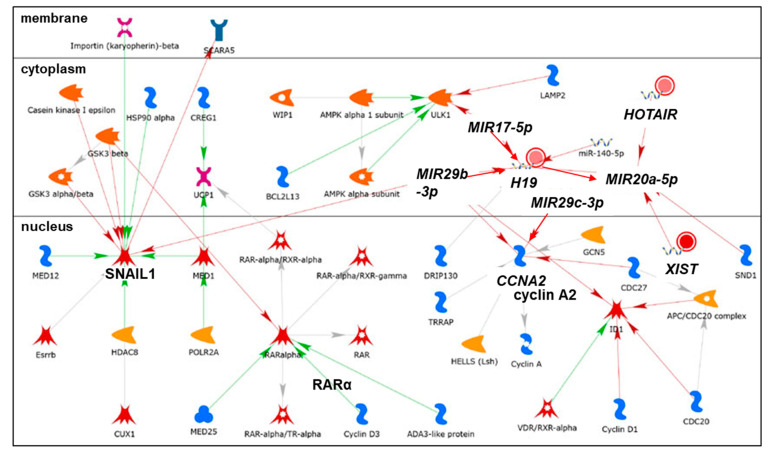
Network 4: *H19, XIST, HOTAIRM1, SNAIL1*, and cyclin A2 (*CCNA1*), identified in lncRNAs differentially expressed in MCF-7 and or LCC9 cells by MetaCore analysis. Green lines with arrows = stimulation; red lines with arrows = inhibition.

**Table 1 cancers-13-03530-t001:** lncRNAs downregulated by miR-29b-1/a in MCF-7 and/or LCC9 breast cancer cells. MCF-7 or LCC9 cells were transfected with pre-miR-control, pre-miR-29b-1-3p, pre-mir-29a-3p, anti-miR-29a (which targets miR-29b-1/a), or anti-miR negative control for a 48 h prior to RNA isolation and RNA sequencing [35]. Values are fragments per kilobase of transcript per million mapped reads (FPKM) and are the average of five replicate samples (GSE81620). Significance was q ≤ 0.05. ND = not detected.

Ensembl	Name	Alias	MCF-7 Pre-miR-29b-1-3p	MCF-7 Pre-miR-29a-3p	MCF-7 AS miR-29a	Signific.	LCC9 Pre-miR-29b-1-3p	LCC9 Pre-miR-29a-3p	LCC9 AS-miR-29a-3p	Signific.	Signific. Different between MCF-7 vs. LCC9	Role in Breast Cancer	Network Supp. Table 2, Figure 2, Figure 3 and Figure 4	miR-29-lncRNA Interaction from DIANA-LncBase v.3
ENSG00000270816	*LINC00221*		ND	ND	ND		4.05	4.26	5.76	yes	LCC9 > MCF-7			no results found
ENSG00000242808	*SOX2-OT*		0.14	0.40	0.37	yes	0.23	0.29	0.15	no	MCF-7 > LCC9	Oncogenic	3	yes, high confidence
ENSG00000223414	*LINC00473*		0.17	0.24	0.11	no	0.04	0.02	0.79	yes	MCF-7 > LCC9	Oncogenic		no results found
ENSG00000215417	*MIR17HG*		1.58	1.69	8.16	yes	1.11	0.87	1.82	no	MCF-7 > LCC9	Oncogenic	7	no results found
ENSG00000213468	*FIRRE*	LINC01200	1.92	1.84	1.55	no	0.51	0.68	0.77	yes	MCF-7 > LCC9			yes, high confidence
ENSG00000215386	*MIR99AHG*	LINC00478	3.61	3.62	5.09	no	6.68	5.91	10.77	yes	LCC9 > MCF-7	Oncogenic		yes, high confidence
ENSG00000227036	*LINC00511*		5.48	5.17	7.83	yes	0.20	0.33	1.01	yes				yes, high confidence
ENSG00000176124	*DLEU1*		15	15	18	no	9.83	11.96	21.06	yes	MCF-7 > LCC9		4	yes, high confidence
ENSG00000214293	*APTR*	RSBN1L-AS1	22	22	19	no	22.87	20.50	27.18	yes		oncogenic	2	no results found
ENSG00000247556	*OIP5-AS1*	Cyrano	26	28	33	no	20.10	22.11	74.54	yes	MCF-7 > LCC9	oncogenic		no results found
ENSG00000225470	*JPX*		29	35	39	no	22.40	20.37	38.48	yes	MCF-7 > LCC9			yes, high confidence
ENSG00000226950	*DANCR*		29	28	29	no	23.93	17.90	75.62	yes	MCF-7 > LCC9	oncogenic	2	no results found
ENSG00000222041	*CYTOR*	LINC00152	37	36	47	no	18.53	15.90	28.57	yes	MCF-7 > LCC9	oncogenic	3	no results found
ENSG00000245694	*CRNDE*		44	49	56	no	43.36	41.71	58.03	yes			2	yes, high confidence
ENSG00000253352	*TUG1*		114	126	173	yes	50.84	58.46	79.56	yes	MCF-7 > LCC9	oncogenic	2, 3	yes, high confidence
ENSG00000269893	*SNHG8*		282	293	245	no	249.11	202.91	340.90	yes	MCF-7 > LCC9			no results found
ENSG00000203875	*SNHG5*		391	411	439	no	218.59	184.31	324.41	yes	MCF-7 > LCC9	oncogenic	2, 6	yes, high confidence
ENSG00000235123	*DSCAM-AS1*		1213	1444	1280	no	140.59	153.88	483.25	yes	MCF-7 > LCC9	oncogenic		no results found
ENSG00000234741	*GAS5*		2052	2000	1822	no	836.63	827.36	1239.31	yes	MCF-7 > LCC9	tumor suppressor	7	yes, high confidence

**Table 2 cancers-13-03530-t002:** lncRNAs upregulated by miR-29b-1-3p andmiR-29a-3p in MCF-7 and/or LCC9 breast cancer cells. MCF-7 or LCC9 cells were transfected with pre-miR-control, pre-miR-29b-1-3p, pre-miR-29a-3p, anti-miR-29a (which targets miR-29b-1/a), or anti-miR negative control for a 48 h prior to RNA isolation and RNA sequencing [35]. Values (FPKM) are the average of five replicate samples (GSE81620). Significance was q ≤ 0.05. ND = not detected.

Ensembl	Name	Alias	MCF-7 Pre-miR-29b-1-3p	MCF-7 Pre-miR-29a-3p	MCF-7 AS miR-29a	Signific.	LCC9 Pre-miR-29b-1-3p	LCC9 Pre-miR-29a-3p	LCC9 AS-miR-29a	Signific.	Signific. Different between MCF-7 vs. LCC9	Role in Breast Cancer	Network Supp. Table 3	miR-29-lncRNA Interaction from DIANA-LncBase v.3
ENSG00000237517	*DGCR5*	NCRNA00037, LINC00037	0.60	0.42	0.38	no	0.69	0.70	0.32	yes		oncogene		yes, high confidence
ENSG00000236824	*BCYRN1*	BC200a, LINC00004	0.79	0.70	0.30	yes	1.76	2.36	2.93	yes	LCC9 > MCF-7	oncogene		no results
ENSG00000214049	*UCA1*		1.32	1.41	0.33	yes	2.06	1.56	1.03	no	LCC9 > MCF-7	oncogene	1,3	no results
ENSG00000225969	*ABHD11-AS1*	LINC00035	2.57	1.57	1.14	yes	0.78	0.79	0.37	no	MCF-7 > LCC9	unknown	2	no results
ENSG00000237886	*NALT1*	RP11-611D20.2	7.77	5.20	3.88	yes	4.51	4.78	2.66	yes	MCF-7 > LCC9	unknown		no results
ENSG00000223573	*TINCR*	LINC00036	17	17	17	no	2.74	2.88	1.77	yes	MCF-7 > LCC9	oncogene		no results
ENSG00000253716	*MINCR*	RP13-582O9.5, LINC01604	30	27	18	yes	20	13	17	no	MCF-7 > LCC9	unknown		yes, high confidence
ENSG00000245532	*NEAT1*		395	372	186	yes	147	143	59	yes	MCF-7 > LCC9	oncogene	1,3	no results
ENSG00000251562	*MALAT1*	NEAT2, LINC00047	496	442	469	no	76	87	54	yes	MCF-7 > LCC9	oncogene	1,3	yes, high confidence

**Table 3 cancers-13-03530-t003:** lncRNAs differentially expressed in MCF-7 and LCC9 breast cancer cells that are not regulated by miR-29b-1/a. Values (FPKM) are the average of five replicates.

Ensembl	Gene	Alias	MCF-7 Avg	Sem	LCC9 Avg	Sem	Signific. Different between MCF-7 vs. LCC9	Role in Breast Cancer
ENSG00000227418	*PCGEM1*	LINC00071	1.18	0.16	0.00	0.00	MCF-7 > LCC9	unknown
ENSG00000234753	*FOXP4-AS1*	RP11-328M4.2	1.58	0.14	2.69	0.60	LCC9 > MCF-7	unknown
ENSG00000130600	*H19*	LINC00008	1.28	0.09	3.56	0.19	LCC9 > MCF-7	oncogenic
ENSG00000251018	*HMMR-AS1*	RP11-80G7.1	1.57	0.09	3.24	0.22	LCC9 > MCF-7	oncogenic
ENSG00000230798	*FOXD3-AS1*	RP4-792G4.2	1.56	0.05	2.45	0.38	LCC9 > MCF-7	oncogenic
ENSG00000257671	*KRT7-AS*	RP3-416H24.1	1.64	0.17	0.98	0.10	MCF-7 > LCC9	oncogenic-metastasis
ENSG00000257557	*PPP1R12A-AS1*	RP11-84G21.1	1.62	0.23	2.84	0.54	LCC9 > MCF-7	unknown
ENSG00000163364	*LINC01116*	AC017048.3	2.55	0.20	4.07	0.16	LCC9 > MCF-7	oncogenic
ENSG00000225953	*SATB2-AS1*		2.29	0.20	1.09	0.61	MCF-7 > LCC9	unknown
ENSG00000224189	*HAGLR*	HOXD-AS1	3.19	0.13	1.95	0.26	MCF-7 > LCC9	unknown
ENSG00000228630	*HOTAIR*		3.58	0.14	9.04	0.50	LCC9 > MCF-7	oncogenic
ENSG00000231133	*HAR1B*		4.31	0.07	0.16	0.04	MCF-7 > LCC9	unknown
ENSG00000236404	*VLDLR-AS1*	RP11-125B21.2	5.85	0.38	1.69	0.10	MCF-7 > LCC9	unknown
ENSG00000237036	*ZEB1-AS1*		4.72	0.52	2.21	0.28	MCF-7 > LCC9	oncogenic
ENSG00000230590	*FTX*		5.75	0.34	2.41	0.04	MCF-7 > LCC9	oncogenic
ENSG00000240498	*CDKN2B-AS1*	ANRIL	6.15	0.23	2.43	0.11	MCF-7 > LCC9	oncogenic
ENSG00000228288	*PCAT6*		5.80	0.05	1.87	0.22	MCF-7 > LCC9	oncogenic
ENSG00000231607	*DLEU2*		5.71	0.47	8.45	0.81	LCC9 > MCF-7	oncogenic
ENSG00000233429	*HOTAIRM1*		8.14	0.35	2.00	0.07	MCF-7 > LCC9	
ENSG00000186594	*MIR22HG*		11.59	1.64	7.42	0.97	MCF-7 > LCC9	tumor suppressor
ENSG00000231721	*LINC-PINT*	AC058791.2	9.15	1.38	3.70	0.68	MCF-7 > LCC9	recurrence
ENSG00000198496	*NBR2*		12.17	1.06	5.39	0.14	MCF-7 > LCC9	tumor suppressor
ENSG00000247828	*TMEM161B-AS1*		12.75	0.72	9.75	1.05	MCF-7 > LCC9	
ENSG00000225978	*HAR1A*		9.05	2.16	0.27	0.26	MCF-7 > LCC9	oncogenic
ENSG00000223749	*MIR503HG*		15.10	0.41	7.22	0.26	MCF-7 > LCC9	tumor suppressor
ENSG00000243410	*PSMD6-AS1*	RP11-245J9.4	19.54	1.72	3.01	0.13	MCF-7 > LCC9	unknown
ENSG00000218537	*MIF-AS1*	AP000350.4	17.53	3.08	106.05	13.72	LCC9 > MCF-7	Oncogenic
ENSG00000182165	*TP53TG1*	LINC00096	26.63	1.06	86.63	4.17	LCC9 > MCF-7	
ENSG00000197775	*DHRS4-AS1*		30.69	2.09	15.33	0.81	MCF-7 > LCC9	tumor suppressor
ENSG00000236901	*MIR600HG*		38.30	1.75	4.64	1.57	MCF-7 > LCC9	tumor suppressor
ENSG00000260032	*NORAD*	LINC00657 LOC647979 TP53TG1	65.34	3.03	44.08	0.97	MCF-7 > LCC9	oncogenic
ENSG00000229807	*XIST*		82.54	6.05	0.01	0.01	MCF-7 > LCC9	tumor suppressor
ENSG00000249859	*PVT1*		107.75	9.24	79.45	6.90	MCF-7 > LCC9	oncogenic
ENSG00000255717	*SNHG1*	LINC00057, U22HG UHG, lncRNA16	581.99	16.98	357.14	41.34	MCF-7 > LCC9	oncogenic
ENSG00000177410	*ZFAS1*		928.45	30.95	282.98	27.79	MCF-7 > LCC9	tumor suppressor

## Data Availability

GEO accession # GSE81620.

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
