# Peer review of "Identification and Roles of miR-29b-1-3p and miR29a-3p-Regulated and Non-Regulated lncRNAs in Endocrine-Sensitive and Resistant Breast Cancer Cells"

_cancers, 2021, doi:10.3390/cancers13143530_

Round 1

Reviewer 1 Report

In the submitted manuscript, which combine bioinformatic analysis of previous results and literature review, Muluhngwi and Klinge have tried to identify miR-29-regulated and -non-regulated lncRNAs and describe their roles in endocrine-sensitive and resistant breast cancer cells (MCF-7 and MCF7/LCC9). Although their literature review is very extensive, their bioinformatic analyses are pretty simplistic and straightforward.

1) There are lots of inconsistencies in writing style, lots of unnecessary spaces exist, abbreviations were used without explanation, scientific jargon phrases were used, several sentences are incomprehensible, gene symbols were inconsistently written in italics, etc. All because of which this manuscript seems very sloppy. Authors should thoroughly check the text and correct all errors and inconsistencies.

2) Authors were inconsistently and erroneously writing names of those four miRNAs they have studied. Since there is an overlap in names of the miRNAs from miR-29 family, to avoid ambiguities, authors should provide miRBase accession number for each of that four miRNAs. Also, miR-29 family has 4, not 3 members: 'a', 'b-1', 'b-2' and 'c'. 'b-1' and 'b-2' cannot be denoted just as 'b', since authors properly stated they are derived from different chromosomes.

3) Line 80: Authors should provide the date on which GENCODE web page were accessed, because current data differ from their! Also, authors should explain more clearly and precisely to what those numbers refer to: all genes or just protein coding ones, genes for lncRNAs (not no. of transcripts), genes for microRNAs or mature microRNA transcipts, etc.

4) Lots of general statements related to microRNAs or lncRNAs could be cited by a single reference. Also, some references seem inappropriate, e.g., in Ref. 38 there is nowhere mentioned the definition of lncRNAs!

5) Lines 95-96: Part of sentence "...or in cis to regulate neighboring genes in chromatin to regulate host or nearby genes" is unclear.

6) Because of its length, sentence in lines 103-105 is also unclear.

7) "Materials and methods" are completely misleading, authors haven't performed any new experiment for this study! Since they used their old data (GEO GSE81620), there is really no point to describe experimental methods at all. It would be more informative if authors have described in (much) more detail how RNA-Seq data were analyzed, and what MetaCore results in general present.

8) Authors should provide URL and reference(s) for MetaCore.

9) Authors should state somewhere clearly that LCC9 cells are just derivation of MCF-7. Someone could think that those are two different types of breast cancer cell lines.

10) Figure 1 in not very comprehensive.

11) Almost all figures from MetaCore are too small and unreadable.

12) There are lots of inconsistencies in writing table elements: mixing capital and small letters in column names and cell items, column names are too long, most abbreviations in tables lack explanation, etc.

What does the sentence "Values (FPKM) are the average of five replicate samples and are FPKM (GSE81620)." in Table 1 and 2 legends mean?

What blue and red fonts in Supp. Table 5 present?

Reviewer 2 Report

I found this paper interesting, and authors did valuable work. However, I would like to address a few comments for authors:

  1. First of all , paper, however valuable is too long. In my opinion some points are described too widely, thus I suggest to reduce Introduction, methods, results and discussion section.
  2. I am a bit confused by the way of data presentation. I think that common division of paragraph into results as well as discussion will be enough.
  3. At least for miRNA-29 the GO and KEGG analysis should be conducted.

Round 2

Reviewer 1 Report

Authors have substantially improved the quality of this manuscript and satisfactorily responded to all my questions and concerns.